# The anticipation of events in time

Matthias Grabenhorst[1,3]*, Georgios Michalareas[1,3], Laurence T. Maloney[2] & David Poeppel[1,2]

Humans anticipate events signaled by sensory cues. It is commonly assumed that two uncertainty parameters modulate the brain's capacity to predict: the hazard rate (HR) of event probability and the uncertainty in time estimation which increases with elapsed time. We investigate both assumptions by presenting event probability density functions (PDFs) in each of three sensory modalities. We show that perceptual systems use the reciprocal PDF and not the HR to model event probability density. We also demonstrate that temporal uncertainty does not necessarily grow with elapsed time but can also diminish, depending on the event PDF. Previous research identified neuronal activity related to event probability in multiple levels of the cortical hierarchy (*sensory (V4)*, *association (LIP)*, *motor* and other areas) proposing the HR as an elementary neuronal computation. Our results—consistent across vision, audition, and somatosensation—suggest that the neurobiological implementation of event anticipation is based on a different, simpler and more stable computation than HR: the reciprocal PDF of events in time.

[1] Neuroscience Department, Max-Planck-Institute for Empirical Aesthetics, Grüneburgweg 14, 60322 Frankfurt, Germany. [2] Department of Psychology, Center for Neural Science, 6 Washington Place, New York, NY 10003, USA. [3] These authors contributed equally: Matthias Grabenhorst, Georgios Michalareas. *email: m.g@ae.mpg.de

Successful perceptual anticipation at the second scale allows organisms to prepare responses before events occur (Fig. 1a). At one extreme, prediction of events may fail entirely: events arrive unexpectedly. At the other, sensory information serves only to confirm what was effectively already known based on accurate prediction, for example in the context of complete stimulus regularity. In between these extremes, information about the likely time of arrival of an event can be summarized as a probability density function (PDF) $f(t)$ across time. For the brain, the estimation of event occurrence is influenced by two main sources of uncertainty: the actual probability distribution of events and the brain's inherent uncertainty in estimating elapsed time[1,2]. In previous work, ranging from single-cell recordings[3–7] to non-invasive electrophysiology[8,9] and neuroimaging[10–12], two compelling hypotheses were advanced for the computations involved in the neural representation of both sources of uncertainty. The two hypotheses are not mutually exclusive and we will examine both.

On Hypothesis A the brain models the probability distribution of event occurrence by computing the hazard rate (HR)[3–10,12–14], an intuitively pleasing and conceptually straightforward model of anticipation. The HR $h(t)$ is defined as the probability density of an event at any point in time $t$, given that it has not occurred before[15]: $h(t) = \frac{f(t)}{1-F(t)}$ where $F(t) = \int_{-\infty}^{t} f(u)\,du$ is the cumulative distribution function (CDF), the probability that the event has occurred at or before time $t$. Although the HR appears to represent precisely the information needed in temporal anticipation, one should be cautious in asserting its validity based on this conjecture alone[16]. One significant challenge for Hypothesis A is that the computation of HR, which requires integration of event probability over time, is relatively complex as well as numerically unstable[15,17]. Technically, the HR is the PDF divided by the survival function $1-F(t)$[15]. Alternatively, the HR can also be interpreted as the PDF multiplied by $\frac{1}{1-F(t)}$, rendering the reciprocal survival function a time-varying scaling factor for the PDF. This suggests that the fundamental variable for HR is

the PDF. Therefore, although the HR and the PDF are typically thought of as separate representations of probability density, each may possibly be important in the brain's effort to model event probability.

Hypothesis A further posits an inverse relationship between reaction time (RT) and event expectation: the RT to an event is linearly anti-correlated with HR[5,9,12] or, put differently, linearly correlated with the mirrored HR (reflected around a constant value).

On Hypothesis B the brain's uncertainty in estimating elapsed time increases monotonically with time, which is often referred to as the "scalar property" of time estimation[18,19]. At the behavioral[5,6,12,14] and neural levels[3,5–7,12,20], this uncertainty is typically modeled as a Gaussian function whose standard deviation increases linearly with elapsed time. The rate of this linear increase is determined by the Weber fraction of time estimation[19]. This linearly increasing blurring is termed here temporal blurring. Critically, temporal blurring ignores potential effects of environmental temporal statistics on time estimation, raising the question whether the brain's estimation of time is influenced directly by event probability.

We emphasize that the two hypotheses, A and B, are not independent. The probability distribution itself is blurred because of the uncertainty in elapsed time estimation, which in turn leads to a blurred hazard rate. Consequently, it should be noted that hypotheses A and B are not tested here against each other but rather are evaluated as model components in temporal anticipation which we investigate at the behavioral level[21].

The combination of hypotheses A and B leads to the mapping rule, relating the brain's temporal-probabilistic input to its RT output (Fig. 1b). Guided by Hypotheses A and B, we tested several such mapping rules by constructing explanatory variables for modeling RT to probabilistically distributed events (Fig. 1c). These variables were derived from either of two basic parameters, namely the HR (Hypothesis A) or the PDF itself. The PDF was selected as an alternative to the HR as it is a fundamental parameter of event probability in time. In a first transformation, each

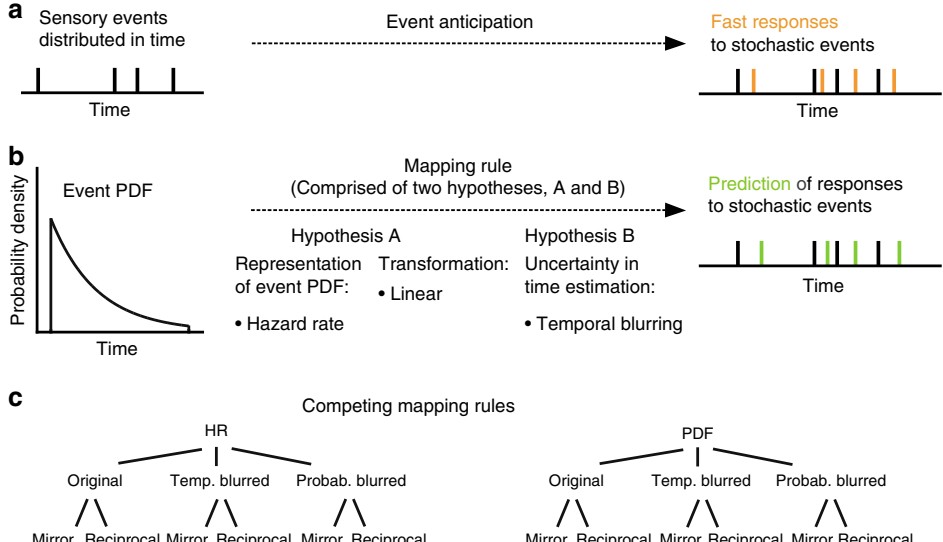

**Fig. 1 Hypotheses and models of event anticipation. a** Event anticipation facilitates fast responses to stochastic sensory events. **b** A mapping rule generates a prediction of reaction time (RT) based on the probability density function (PDF) of events. Mapping rule constituents include a representation of event probability, a transformation function between event probability and RTs, and a model of uncertainty of time estimation. **c** Overview of variables for temporal anticipation derived from mapping rules. The canonical hazard rate (HR) model is based on three assumptions: (1) the brain employs the HR to anticipate events in time. (2) Time estimation contains uncertainty that scales linearly with elapsed time (temporal blurring). (3) RT to events and the HR are linearly anti-correlated (mirror). The three assumptions are investigated using additional, PDF-based variables incorporating probabilistic blurring and a non-linear, reciprocal relationship between RT and model (see Methods).

basic parameter was subjected to one of two different types of blurring, reflecting uncertainty in the estimation of elapsed time. In the first type, temporal blurring (Hypothesis B), the variance of the Gaussian uncertainty kernel increases linearly with elapsed time. In the second type, termed probabilistic blurring, the variance of the kernel depends on the probability distribution of events across time (see Methods). The original, non-blurred, parameters were also tested in comparison to the blurred ones. Given the inverse relationship between reaction times and HR (Hypothesis A), in a second transformation all variables resulting from transformation one, were subjected to one of two types of inversion: either a linear transformation ("mirror") which corresponds to mirroring a variable around its mean (Hypothesis A), or a simple non-linear transformation, the reciprocal of a variable (see Methods). This reciprocal relationship between RT and event probability implicates a non-linear downweighting of low probability events (resulting in relatively longer RTs) relative to an upweighting of high probability events (leading to relatively shorter RTs), which we hypothesize may reflect a more economic deployment of attention in time, benefitting the anticipation of event occurrence. Another aspect making the reciprocal PDF be a more plausible candidate for learning probabilities is its close resemblance to "suprisal" from Shannon's information theory. "Surprisal" is defined as the Shannon information of an outcome, because it represents the amount of surprise when the outcome has been observed[22]. Shannon information is defined as the logarithm of the reciprocal of outcome probability, $\log_2\left(\frac{1}{p(x)}\right)$ bits, where $p(x)$ is the probability of the outcome x. The reciprocal PDF of an event is closely related and can actually be transformed into the outcome probability $p(x)$.

In a "set-go" paradigm (Fig. 2a), participants were asked to respond as fast as possible to the 'go' cue. To examine modality-specific and modality-independent aspects of event anticipation, the stimuli were presented separately in three sensory modalities (vision, audition, and somatosensation). The 'go' time (i.e. between 'set' and 'go') was sampled from one of two different probability distributions, an exponential ($PD_{Exp}$) and its "flipped" version ($PD_{Flip}$). These two 'go' time PDFs (Fig. 2b, black lines) were selected because they are symmetric to each other, while, critically, their HRs are not (Fig. 2c, black lines). This feature allowed for differential investigation of PDF and HR as models of RT. For each distribution, 12 mapping rules were employed; consequently 12 explanatory variables were derived to model RT as a function of 'go' time. (Fig. 1c).

## Results

**Hazard-rate-based models of RT.** In all, 24 subjects generated ~3500 RTs each. In all three modalities, the RTs were strongly modulated by the 'go' time probability distributions. Specifically, the two symmetric PDFs presented, $PD_{Exp}$ and $PD_{Flip}$, lead to nearly symmetric patterns of RT. All explanatory variables were derived from the HR and PDF of each of the presented 'go' time distributions. First, we fitted the RT data with a linear model of the commonly proposed variable "temporally-blurred, mirrored HR". This popular model failed to fit the data adequately. In the $PD_{Exp}$ condition (Fig. 3a), the explanatory variable captured the behavior of the RT data mostly in the latter half of 'go' times. In the early part of the distribution, there were significant deviations between data and explanatory variable, reflected in low $R^2$ values of the fitted models.

In the $PD_{Flip}$ condition (Fig. 3b), the deviation of the explanatory variable from the observed RTs was striking in all modalities, suggesting that Hypotheses A and $B$ do not hold as canonical rules across different probability distributions. The results indicate that the mirrored HR is not employed by the

brain to model event probability across time. We next tested whether the inverse relation between RT and HR can be better captured by a simple non-linear transformation. To do this, we replaced the mirrored transformation with the reciprocal, in which the HR is inverted by division instead of being linearly mirrored. This variable, the "temporally-blurred, reciprocal HR", significantly improved the fit to RT in the $PD_{Flip}$ condition in all modalities (Fig. 3d). However, in the $PD_{Exp}$ condition we observed no improvement but rather a deterioration in the fit (Fig. 3c). Thus, the HR is an unlikely transformation to model event probability across time.

**PDF-based models of RT.** We next turned to models based on a more fundamental, 'core' probability parameter, the PDF itself. Similar to HR, we first examined the variable "temporally-blurred, mirrored PDF". Although the shape of the explanatory variable captured trends in the data, the fit to RT was poor, especially in the $PD_{Exp}$ condition (Supplementary Fig. 3c). We then examined the non-linear, reciprocal transformation by using the "temporally-blurred, reciprocal PDF" as an explanatory variable (Fig. 4a, b).

The fit to the data was significantly improved, as shown by the similar behavior of model and data across 'go' time, confirmed by the high $R^2$ values. Evidently, the reciprocal transformation of the PDF provided a much better fit than the mirror transformation as well as both transformations of the HR. In the $PD_{Flip}$ condition in all modalities, we observed that the explanatory variable settles to a plateau of values quite early, at a 'go' time of around 1 s, while the actual data continue to have a negative slope (Fig. 4b, gray shading). Additionally, in the early range of 'go' times (e.g. 0.5–0.9 s) the explanatory variable decreases with a steeper slope than the data. These systematic differences between model and data were introduced by the temporal blurring. Therefore, in the early range of 'go' times the Gaussian kernel has a small variance and causes only little blurring, less than the data suggest. In the late range of 'go' times, the Gaussian kernel has a large variance, leading to stronger blurring of the explanatory variable, much more than the data suggest.

**Event probability modulates elapsed time estimation.** These observations lead us to question the concept of monotonically increasing uncertainty that scales with elapsed time itself. Instead, we hypothesize that uncertainty in elapsed time estimation is modulated by temporal probability. This results in a blurring kernel with large variance in the early, less probable range of 'go' times in the $PD_{Flip}$ condition and in a blurring kernel with small variance in the late, more probable range of 'go' times. We applied this probabilistic blurring to the reciprocal PDF and fitted this variable to RT. This simple operation drastically improved the fit in the $PD_{Flip}$ condition (Fig. 4d). In the $PD_{Exp}$ condition (Fig. 4c), the fit was almost identical to the temporal blurring case (Fig. 4a). This is expected, as in both blurring cases the variance of the Gaussian kernel monotonically increased with the magnitude of 'go' times. To quantify differences in model fit between temporally and probabilistically blurred variables, we divided the range of 'go' times into four equal-sized bins. Within each bin, the sum of squared residuals was calculated for the "temporally-" and "probabilistically-blurred, reciprocal PDF". In the $PD_{Exp}$ condition (Fig. 4e) the quotient is relatively close to 1 in all bins, indicating residuals of similar magnitude across the two models and thus a similar goodness-of-fit. In the $PD_{Flip}$ condition (Fig. 4f) the quotient is generally larger, and always higher than 1, which shows that the probabilistically blurred model yielded smaller residuals in all bins and in all modalities. Clearly, probabilistic blurring represented the uncertainty in elapsed time estimation better than

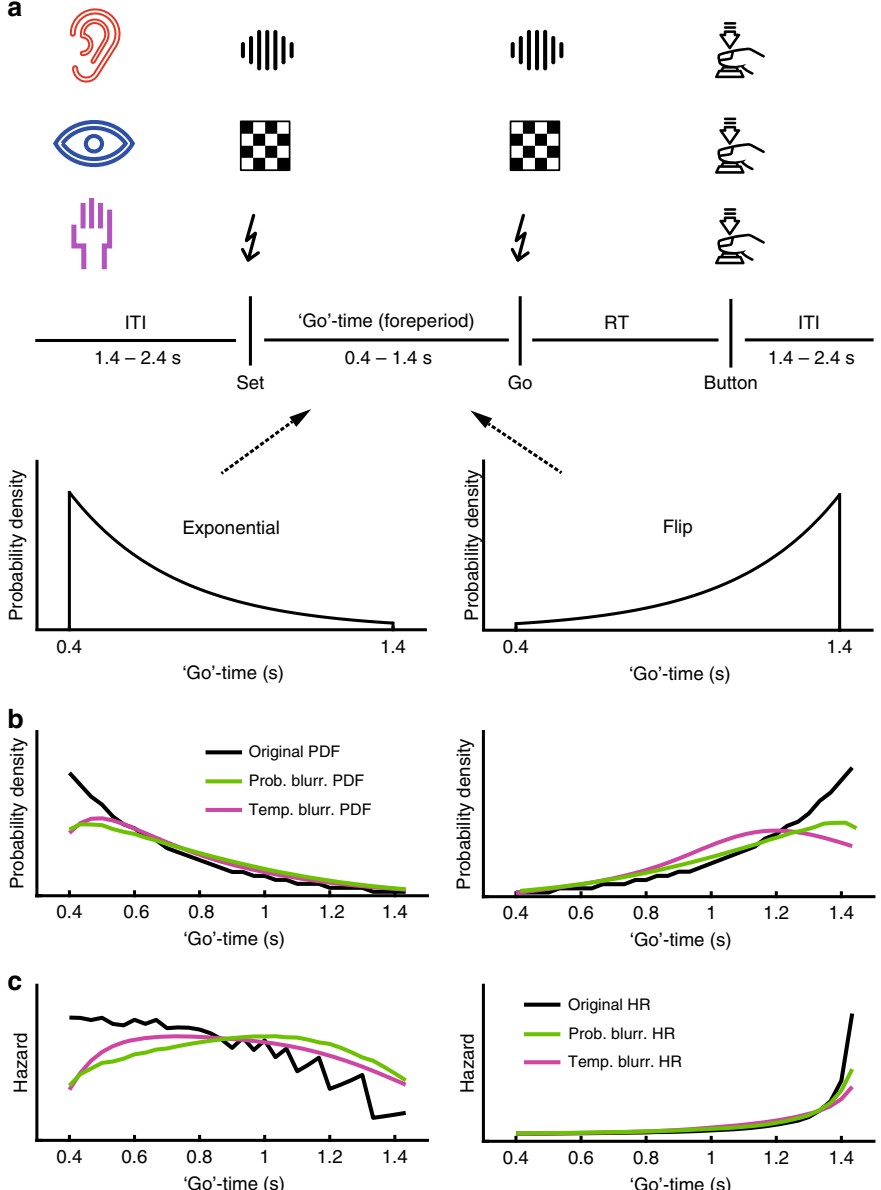

**Fig. 2 Task and presented probabilistic structures. a** Schematic of 'set' - 'go' task. In auditory, visual, and somatosensory blocks of 'set' - 'go' trials, subjects were asked to respond as fast as possible to the 'go' cue by pressing a button with their right index finger. In 9.09% of trials, no 'go' cue was presented (catch trials) in which case subjects were asked not to respond. The time between 'set' and 'go' (the 'go' time) varied systematically. Within a block of trials the 'go' time was drawn from either a truncated exponential distribution (PD$_{Exp}$) or its left-right flipped counterpart (PD$_{Flip}$). **b** The two presented event PDFs, PD$_{Exp}$ and PD$_{Flip}$, in their original, temporally, and probabilistically blurred versions (see Methods). **c** The hazard rates (HR) of PDF$_{Exp}$ and PDF$_{Flip}$ in their original, temporally, and probabilistically blurred versions (see Methods). ("Press Button Halfway Icon" created by H Alberto Gongora via the Noun Project (thenounproject.com/term/press-button-halfway/637743) licensed as Creative Commons CCBY).

the commonly employed temporal blurring. For comparison and completeness, three more variables, the "probabilistically-blurred, mirrored HR", the "probabilistically-blurred, reciprocal HR" and the "probabilistically-blurred, mirrored PDF" were also investigated (Supplementary Figs. 1–3). None of these variables provided a better fit than the variable "probabilistically-blurred, reciprocal PDF". We conclude that, in all three modalities, the probabilistically-blurred, reciprocal PDF—but not the HR—is the most adequate model for mapping the brain's temporal-probabilistic input onto its output, i.e. reaction times.

**Control analyses.** The above results were further validated by two control experiments and various control analyses. These included split-data analyses and single-subject analyses (Supplementary

Note 1, Supplementary Figs. 4–12), a control experiment without catch trials (Supplementary Note 2, Supplementary Fig. 13), and a control experiment using a Gaussian 'go' time distribution (Supplementary Note 3, Supplementary Fig. 14).

**Modality-specific processes at short 'go' times.** All three distributions (exponential, flipped exponential, and Gaussian, for the latter see Supplementary Fig. 14) show that there is an upward bend in RT curves at short 'go' times in vision and somatosensation. This effect is absent in the case of audition, in which the models fit the RT curves even at short 'go' times. Note that neither the HR-based, nor the PDF-based models contain a parameter for modality-specificity. Given the differences across sensory modalities, a dedicated model component is needed to

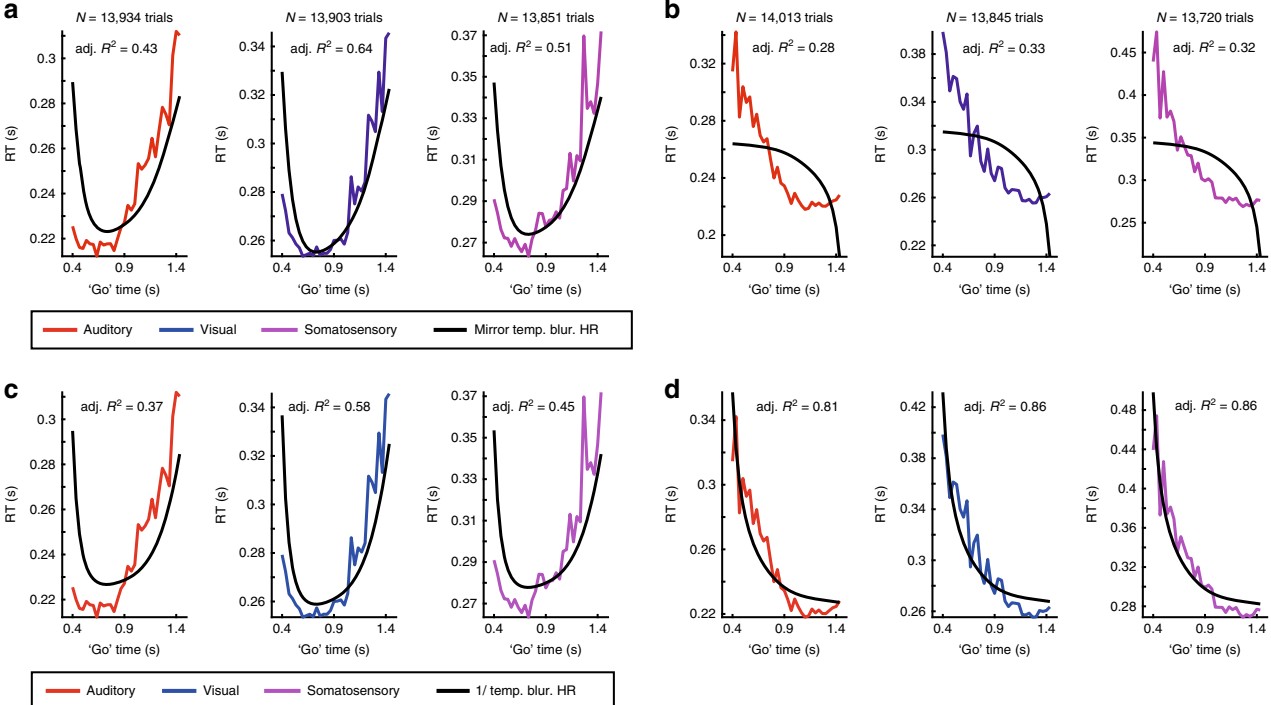

**Fig. 3 Models based on the hazard rate (HR) fail to capture reaction time (RT) data. a** The commonly used temporally-blurred HR (black line) in its linearly transformed ("mirror") version fitted to auditory (red), visual (blue), and somatosensory (violet) RT data from PD$_{Exp}$ condition. The HR model deviates from RT in the early range of 'go' times. **b** As **a** but for PD$_{Flip}$ condition. Note the substantial, qualitative mismatch between the HR model and the RT data. **c** Fits of the temporally-blurred, reciprocal ("1 / ") HR to RT in the PD$_{Exp}$ condition. The fit at the early 'go' times is not improved by the reciprocal model, compare with **a**. **d** As **c** but for PD$_{Flip}$ condition. The reciprocal relationship between HR model and RT improves the fit in the PD$_{Flip}$ condition, compare with **c**. See Supplementary Figs. 1 and 2 for detailed summary of all other fitted HR-based models.

account for modality-specificity. To quantify these effects, we used the auditory condition as a reference and subtracted the auditory RT from visual and somatosensory RT. The difference curves show that the highest ΔRT values occur at the shortest 'go' times and thereafter ΔRT monotonically decreases with 'go' time (Fig. 5a). This pattern was observed in all conditions, regardless of distribution. This confirmed the hypothesis that the processing of events at shorter 'go' times differs between audition and vision/somatosensation. The process leading to the observed ΔRT curves, appears to be captured well by a simple exponential function of 'go' time (Eq. (19), Fig. 5a black fit curves). Adding this fitted exponential model to the PDF-based model from the auditory condition drastically improved the model fit in vision and somatosensation (Fig. 5b). The hypothesized exponential process may be related to differences in time estimation between audition and the other modalities, as audition has been suggested to process events faster[23] and more precisely[24–26]. Alternatively, the modality-specificity in RT might result from differences in the processing of probability over time.

**Modality-specific and modality-independent components of RT.** The striking similarities in processing temporal-probabilistic structures across sensory modalities suggest shared neural processes, while the differences, e.g. in processing speed, likely reflect modality-specificity. The RT distributions are the result of the superposition of such neural mechanisms. Accordingly, we characterize the RT distributions based on the hypothesis of separate, additive contributions to the mapping of an event PDF onto the corresponding RT. One contribution presumably is modality-specific, reflecting more peripheral processing stages, including latencies in sensory signal transduction and feed-

forward information processing[27–34]. The other contribution is hypothesized to reflect modality-independent processes associated more with the processing of the 'go' time PDF itself. QQ-plots revealed overall similarities in RT distribution across PD$_{Exp}$ and PD$_{Flip}$ conditions, but also heavier right tails in the PD$_{Flip}$ condition in all three modalities (Fig. 6a).

We next investigated these distributional differences. Participants' RTs exhibited two characteristic patterns, one modality-specific, the other modality-independent. In the modality-specific pattern, median RT was faster in the auditory condition than in both visual ($-42.4 \pm 26.6$ ms, mean $\pm$ standard deviation, SD, $P = 8.2 \times 10^{-7}$; Tukey's honestly significant difference test) and somatosensory conditions ($-54.8 \pm 33.7$ ms, mean $\pm$ SD, $P = 1.1 \times 10^{-9}$; Tukey) (Fig. 6b). In RT variance, a different modality-specific pattern emerged. Interquartile range (IQR$_{RT}$) was smaller in vision ($-21.3 \pm 26.4$ ms, mean $\pm$ SD, $P = 0.00026$; Tukey) and audition ($-14.6 \pm 26.2$ ms, mean $\pm$ SD, $P = 0.0198$; Tukey) compared to somatosensation (Fig. 6c). In the modality-independent pattern, the change in 'go' time PDF had no effect on median RT (Fig. 6b). However, IQR$_{RT}$ was significantly larger in the PD$_{Flip}$ condition as compared to PD$_{Exp}$ (Fig. 6c). The magnitude of the difference in IQR$_{RT}$ between probabilistic conditions did not differ between sensory modalities ($F_{(2,22)} = 0.04$, $P = 0.96$, one-way ANOVA (for analysis of variance and planned contrasts on median RT and IQR$_{RT}$ see Tables S1–S4).

In Fig. 4a differences between auditory RT, on one side, and visual and somatosensory on the other, are evident at shorter 'go' times. In the visual and somatosensory cases there are characteristic upward bends that are not seen in the auditory modality. To investigate whether these differences in RT curves between modalities drive the above results on median RT and IQR$_{RT}$, the respective analyses were performed again on RT from a subset of

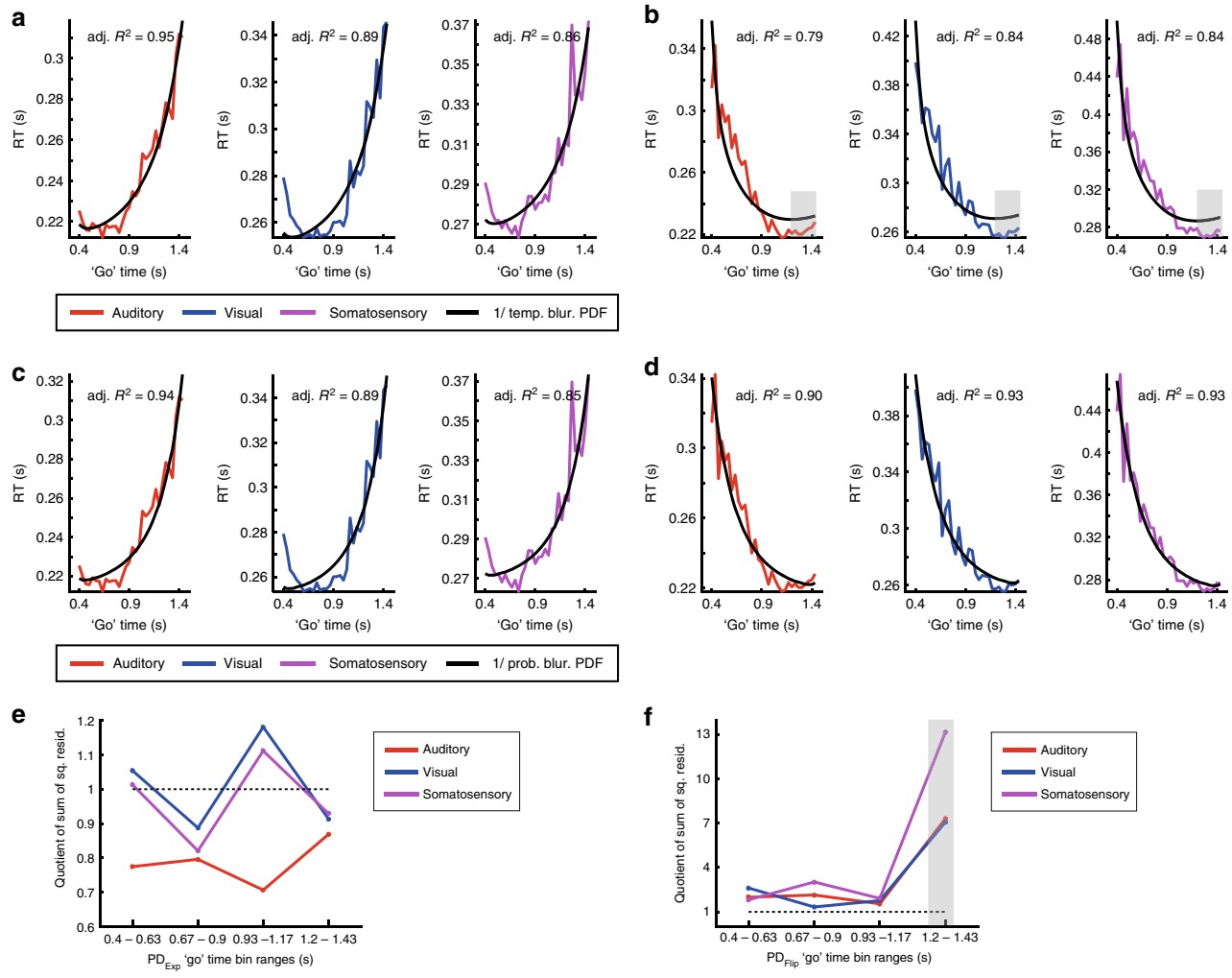

**Fig. 4 Models based on probabilistically-blurred, reciprocal PDF capture reaction time (RT). a** Model fits to auditory (red), visual (blue), and somatosensory (violet) RT based on the temporally-blurred, reciprocal PDF (black line) in the $PD_{Exp}$ condition. **b** As **a** but for $PD_{Flip}$ condition. Since temporal blurring increases with 'go' time, it has a stronger smoothing impact on the model at longer 'go' times. This feature is not reflected in the data ('go' time span 1.2 to 1.43 s highlighted). **c** Model fits to RT based on the probabilistically-blurred, reciprocal PDF in the $PD_{Exp}$ condition, **d** As **c** but for $PD_{Flip}$ condition. **e** Comparison between two reciprocal PDF-based models ("temporally-blurred" and "probabilistically- blurred"), in each of four equally spaced bins across the 'go' time span. **e** In the $PD_{Exp}$ condition, the quotient of squared residuals ("temporally-blurred"/"probabilistically-blurred") is relatively close to 1 in all bins, indicating residuals of similar magnitude across the two models and thus a similar goodness-of-fit. **f** As **e** but for the $PD_{Flip}$ condition, residuals are bigger in the temporally-blurred model compared to the probabilistically-blurred model in all bins, especially in bin #4 (highlighted). See Supplementary Figs. 1 and 3 for a detailed summary of all other fitted PDF-based models.

'go' times (0.5667 s to 1.4 s), thus eliminating the potentially confounding differences in RT curves. The analyses yielded highly similar results to the ones above (Supplementary Fig. 16).

To identify how the systematic differences in RT distribution (Fig. 6a) relate to the observed modality-specific and modality-independent patterns, we modeled the RT distributions with an exponential-Gaussian PDF. This two-process model is the convolution of an exponential PDF (parameter $\tau$) and a Gaussian PDF (parameters $\mu$ and $\sigma$) (Fig. 6d). It states that the generation of RT depends on a sum of peripheral Gaussian processes and a central decision process that is hypothesized to be exponential[15,35]. The model provides an excellent fit to the data in all conditions, both at the group level (Fig. 6e) and at the single-subject level (Supplementary Fig. 15). Both Gaussian parameters were sensitive to changes in sensory modality but were insensitive to changes in the 'go' time PDF (Fig. 6f, h), in line with the model's implicit claim of peripheral Gaussian processes. In contrast, the exponential parameter $\tau$ was *sensitive* to the 'go' time

PDF (Fig. 6g), having larger values in the $PD_{flip}$ condition as compared to $PD_{Exp}$. This pattern of $\tau$ closely mirrors the behavior of $IQR_{RT}$ (Fig. 6c) (See Supplementary Tables 1–4 for analysis of variance and planned contrasts on $\mu$, $\sigma$, and $\tau$). We found that Gaussian $\mu$ captured the behavior of median RT in all three modalities (Supplementary Fig. 17a, b). Likewise, and also independent of modality, $\tau$ captured $IQR_{RT}$ (Supplementary Fig. 17c, d). Taken together, these findings suggest that only the process(es) reflected in the exponential parameter $\tau$ use information on the 'go' time PDF. Taken together, our findings on the influence of the 'go' time PDF on the RT distribution support (i) the broader hypothesis of peripheral and central processes involved in event anticipation as well as (ii) the specific claim of an exponential process shared by all three modalities.

## Discussion

We investigated how the brain infers probability as a function of time based on sensory input, by analyzing RT to temporally

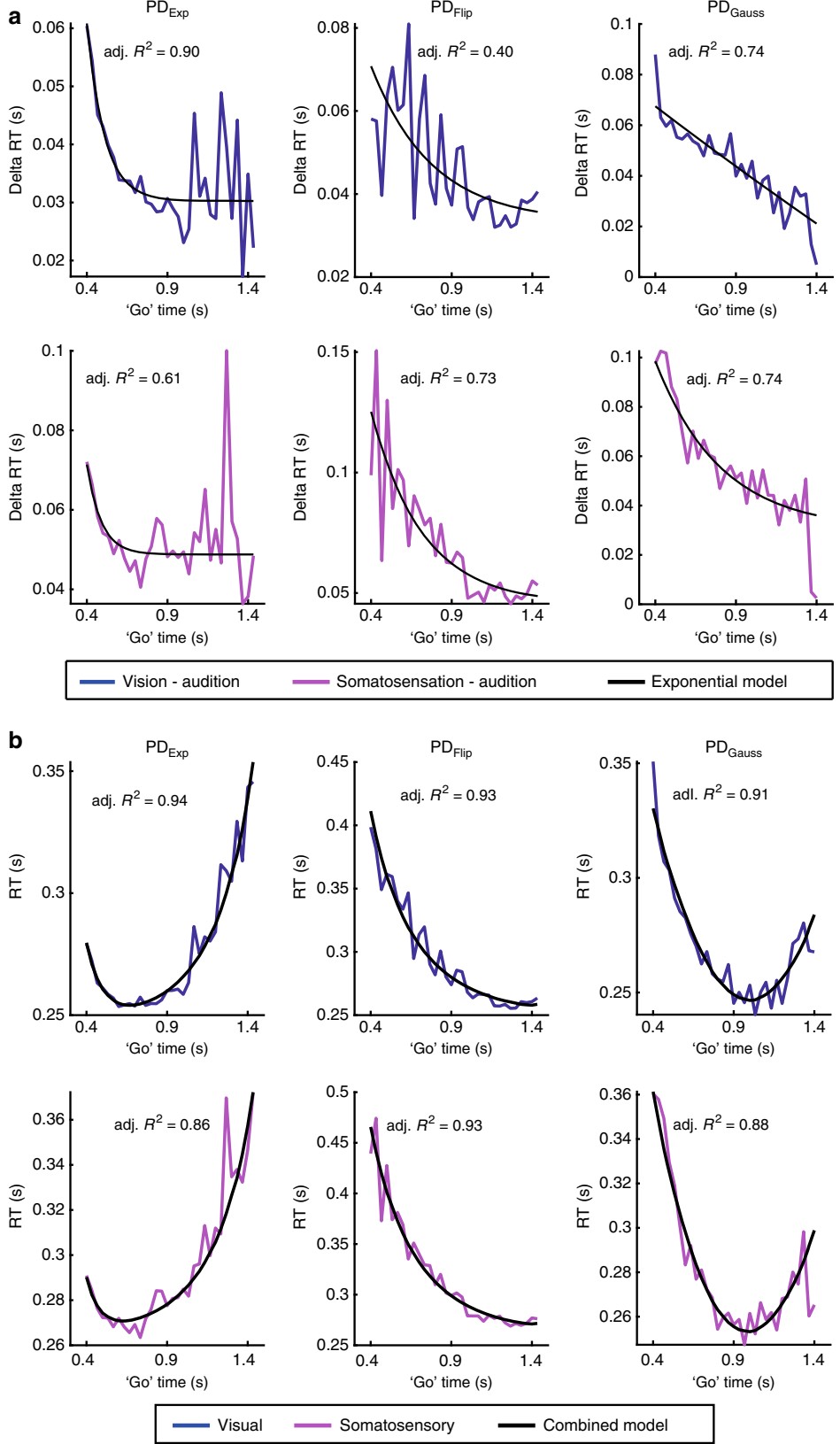

**Fig. 5 Exponential function of 'go' time captures RT differences between modalities. a** Within exponential, flipped exponential and Gaussian conditions, ΔRT curves were calculated by subtracting auditory RT curves from visual (blue) and somatosensory (magenta) ones. The ΔRT curves show a decrease over the range of 'go' times, most pronounced at shorter 'go' times, for all distributions. An exponential model (black curves) of 'go' time (Eq. (19)) captured the ΔRT curves' behavior. **b** The fitted exponential model was added to the reciprocal, probabilistically-blurred PDF model fitted to auditory RT and plotted against visual and somatosensory RT. This combined model captured the previous deviations between data and model at shorter 'go' times in all conditions in both vision and somatosensation.

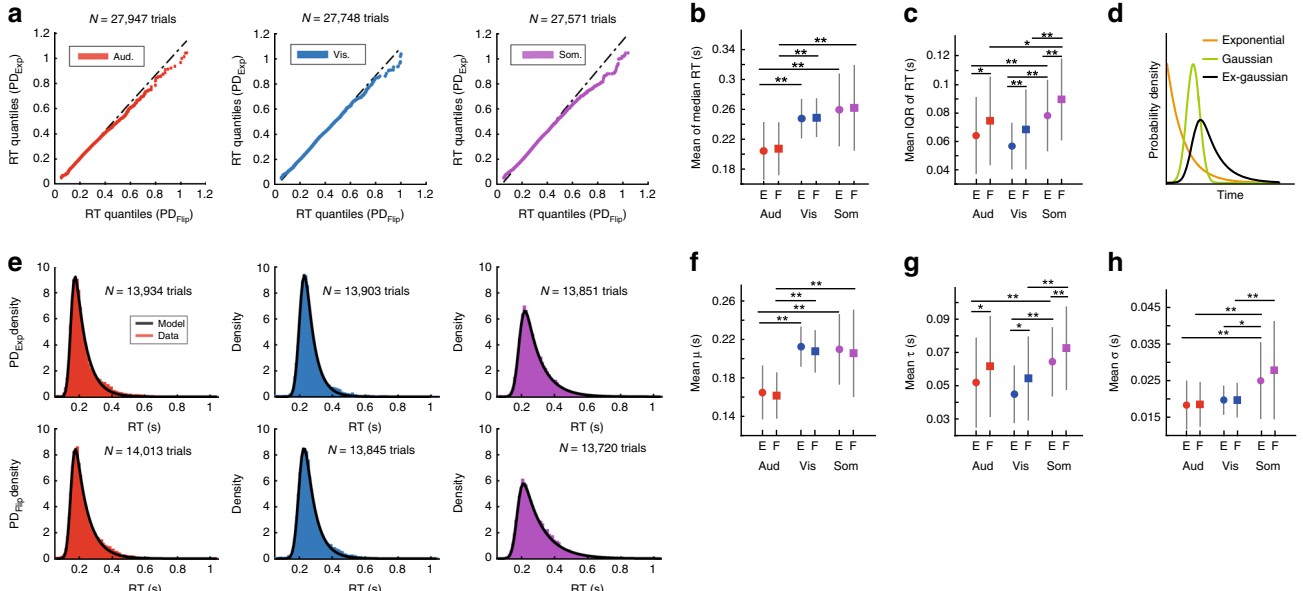

**Fig. 6 Reaction time (RT) variance is sensitive to the probability distribution of events. a** In all three sensory modalities, QQ-Plots reveal heavier right tails in the RT distributions in the $PD_{Flip}$ condition. **b** Mean of median RT in $PD_{Exp}$ ('E') and $PD_{Flip}$ ('F') conditions. Average RT is not sensitive to the presented probability distribution. (planned contrasts, *$P < 0.05$, **$P < 0.01$, two-tailed Student's $t$-test). **c** Mean of interquartile range of RT in $PD_{Exp}$ and $PD_{Flip}$ condition. RT variance is higher in the $PD_{Flip}$ condition (planned contrasts, *$P < 0.05$, **$P < 0.01$, two-tailed Student's $t$-test). **d** The Ex-Gaussian distribution is a convolution of an exponential and a Gaussian PDF. **e** The ex-Gaussian model fits RT histograms on the group level (**e**) and on the single-subject level (see Supplementary Fig. 15). **f–h** Ex-Gaussian fit parameters from single-subject fits. **f** Mean of Gaussian $\mu$ resembles mean of median RT (compare to **b**, planned contrasts, *$P < 0.05$, **$P < 0.01$, two-tailed Student's $t$-test). **g** Mean of exponential $\tau$ resembles interquartile range of RT (compare to **c**, planned contrasts, *$P < 0.05$, **$P < 0.01$, two-tailed Student's $t$-test). **h** Mean of Gaussian $\sigma$ is not sensitive to the presented probability distribution (planned contrasts, *$P < 0.05$, **$P < 0.01$, two-tailed Student's $t$-test). Error bars denote standard deviation. For ANOVAs, see Supplementary Tables 1–4.

distributed events. In audition, vision, and somatosensation, subjects evidently extracted and used the probabilistic information encoded in the trial structure to predict event onsets, as shown by their distinct patterns of RT modulation. Previous work has proposed the hazard rate as a model of probability over time used by the brain to anticipate events and plan responses. We demonstrate that it is not the HR but the PDF—a simpler and more stable variable—which better captures the data.

We found that RT was sensitive to changes in event probability density. Notably, the observed RT modulation in the $PD_{Exp}$ condition indicates that subjects inferred temporal-probabilistic information from exponentially distributed events. This is remarkable because it is commonly assumed that an exponential distribution renders temporal prediction impossible[7,36], arguably due to its flat HR in the case of no catch trials.

We first modeled the RTs using various explanatory variables based on linear and non-linear transformations of the HR. None of these variables adequately captured RT patterns across the different 'go' time distributions. Therefore, the HR is not a likely parameter the brain uses to relate an event PDF to its corresponding RT. Instead, we demonstrate that a non-linear transformation of the PDF, the reciprocal, better captures the RT behavior across different PDFs in three sensory modalities.

The models based on the reciprocal PDF clearly provided the better fits to RT in all conditions as compared to the reciprocal HR. Interestingly, the reciprocal HR is just the reciprocal PDF multiplied by the survival function, as $\frac{1}{HR} = (1 - CDF) \times \frac{1}{PDF}$. In this equation the survival function appears as a time-varying scaling factor of the reciprocal PDF. Following this revealing result that the PDF, and not the HR, is the most likely parameter modeled by the brain, and given that $HR = \frac{1}{survival\ function} \times PDF$, it follows that the brain did not employ the scaling factor (1/survival function) in order to scale the PDF. Instead the scaling

appeared to be better approximated by a fixed value, uniform for the entire range of 'go' times. Nonetheless, it seems possible that in contexts other than simple event anticipation, the brain might use the survival function as a scaling factor for event probability density in which case the HR may be an appropriate parameter of probability in time.

We note that in a reward-based context, the HR has been shown to adequately describe temporal expectation[4–7,10] in a 100% rewarded condition, but not when uncertainty of reward is introduced[36,37]. In the experiments we performed here, no reward was delivered. Therefore, the fact that the PDF, but not the HR, provided the best model of RT might be related to the absence of reward. Although it is clearly beyond the scope of our study to identify how reward modulates anticipation (which includes estimation of event probability and of elapsed time, see below), a simple, intuitive hypothesis can be formulated. If the brain employed a scaling factor for the PDF, which in our case is uniform across time, then under reward conditions this scaling factor could approximate $\frac{1}{survival\ function}$, and the resulting parameter encoded would be the HR, as $HR = \frac{1}{survival\ function} \times PDF$. The survival function is defined as 1—CDF, where the CDF captures the cumulative probability that an event should have happened up to and including the current time instance. The computation of this accumulated probability is cognitively demanding and would likely be facilitated by motivating effects of expected reward. This simple hypothesis could describe a basic mechanism by which the brain incorporates reward into event anticipation (Supplementary Fig. 18a). It suggests a possibility how the PDF-based model we describe here could link to the HR-based models in the reward literature.

Commonly, HR-based models of RT incorporate the concept of temporal blurring[3,5,6,12] which assumes that the uncertainty in elapsed time estimation increases linearly with time[19]: longer

intervals carry higher uncertainty in their estimation than shorter intervals. This implies that the brain's capacity to react fast and accurately to longer timespans is limited compared to shorter timespans, irrespective of the accuracy of the brain's estimate of event probability in time. In other words, the error in time estimation ultimately constrains the brain's benefit from temporal-probabilistic inference. By modeling RT, we found that probabilistically-blurred models fitted the data better than the temporally-blurred ones. In particular, the RTs at the most probable, longer 'go' times in the $PD_{Flip}$ condition were much better captured by the probabilistically-blurred model compared to the temporally-blurred one (Fig. 4b, d). This finding challenges the common assumption that the brain models elapsed time with uncertainty that increases with time per se[2]. Instead it seems that uncertainty in time estimation also scales with probability. We suggest that by modeling its environment's temporal-probabilistic structure, the brain can overcome what has been considered a built-in limitation: the uncertainty in time estimation.

In addition to event probability in time and uncertainty in time estimation, a third source of uncertainty can be quantified in event anticipation: the uncertainty of event occurrence. Suppose a 'go' cue will certainly have occurred by the end of a trial. In this case there is no uncertainty of event occurrence towards the right extremum of the 'go' timespan. In contrast, 'go' cue occurrence may remain uncertain when in a percentage of trials no event occurs (catch trials). The HR has been proposed in both contexts, without catch trials[5–8,10,12] and with catch trials[4,9], as an important model of probability in time. Notably, in a setting with catch trials, certainty of event occurrence is reflected by the CDF asymptotically approaching 1. This leads to a steep increase of the HR's slope towards the end of the 'go' time period, irrespective of probability density (Supplementary Fig. 18b)—the exponential PDF being a rare exception to this. Commonly, this CDF-based up-weighting of 'go' time probability is argued to reflect anticipation, which conforms with intuition as it maximizes event probability towards the right extremum of a timespan when the event will inevitably occur. However, as a result of the CDF approaching 1, the HR's slope becomes very steep and its values approach infinity. Since there is a lower bound to reaction time, this behavior towards the end of a timespan challenges the concept of the HR as a model of RT. Although the commonly employed temporal blurring remedies this problem somewhat by reducing HR values as time increases, the conceptual issue of an ever-increasing variable remains.

Another general problem of the HR concerns its instability in calculation in both contexts with and without catch trials. The HR is difficult to estimate from empirical data because technically, it requires several steps: computation of PDF, integration of PDF to arrive at CDF, transformation of CDF to arrive at the survival function, division of PDF by survival function. Even small errors in the representation of PDF or its CDF will lead to large, unpredictable errors in HR which could have considerable consequences for an organism relying on the HR as its model of temporal probability. The term 1/PDF, on the other hand, can be interpreted as "one-in-many"—a simpler and more stable computation. For example, if the probability of an event is 0.02, the brain might interpret it as "1/0.02", i.e. "one-in-fifty". This interpretation links the 1/PDF model closely to surprisal as defined in Shannon's information theory, $\log_2\left(\frac{1}{p}\right)$ bits, where $p$ is the probability of an event. This suggests that the brain might be computing a simplified version of surprisal. It should also be noted that in some of the previous work on the topic, the observed relation between RT and HR might also have been well captured by the PDF, as both HR and PDF can be monotonically in- or decreasing.

In addition to the influence of event probability, basic timing mechanisms may be central to the anticipatory processes investigated here. The estimation of elapsed time is highly relevant in common interval timing tasks which involve time estimation, production and reproduction[18] and also in the simple RT task presented here. This invites brief discussion of the potential underlying mechanisms. A popular hypothesis proposes two dissociable circuits underlying interval timing, a more "automatic" system that involves the cerebellum and may be used more in the short sub-second range, and another, cognitively controlled system, incorporating the basal ganglia, and related cortical structures that may be involved in processing timespans in the seconds range[18,38]. Our behavioral design does not permit differentiation between neural mechanisms. Still, the differences observed between sensory modalities at shorter 'go' times point towards a related differentiation between components of timing processes: some that are of a more peripheral—arguably "automatic"—nature, and others that are of a more central—eventually more cognitively controlled—nature.

It has been suggested that temporal discrimination is more precise in audition than in vision[24,39–41] and somatosensation[25,42]. Therefore, we used audition as a reference condition, and modeled the across-modality ΔRT curves with an exponential function of 'go' time. These exponential models of ΔRT were added as a component to the PDF-based model which drastically improved the fit, accounting for the modality-specificities. This modeling procedure relies on the use of audition as a reference modality, which naturally raises further questions: is audition itself free of modality-specific effects? Which components of timing are indeed modality-specific and which components are modality-general[43]? These important aspects cannot conclusively be addressed by this experiment. Nonetheless, although we cannot further specify the processes underlying the exponential model component, we hypothesize that the exponential function of 'go' time reflects differences between modalities in fundamental timing processes in the sub-second range. A substantial body of previous work proposed that in interval discrimination for timespans above 0.2 s (up to 1 s)[38] and in a variety of timing tasks between 0.1 and 1.5 s[44], the Weber fraction for time estimation is close to constant. Such evidence from interval discrimination suggests that the modality-specific RT modulation at short 'go' times we observed may be unrelated to elapsed time estimation itself. Instead, the impact of event probability on time estimation appears to be a likely source of modality-specific RT modulation. Given that a large literature on the processing of probability in time investigates the relationship between sensory input and behavioral output, we emphasize the importance of the contingencies of sensory input modalities for any potential inferences made.

To further investigate the influence of sensory input modality on processes involved in temporal-probabilistic inference, we analyzed the RT distributions. We observed modality-specific differences in average RT that are in agreement with existing findings covering a wide range of simple RT and go/no-go tasks[15,23]. Here they are seen in the context of event anticipation.

The RT distributions could be decomposed into the sum of a modality-independent PDF which was exponential and a second, modality-dependent PDF which was Gaussian. The Gaussian parameter $\mu$ corresponded to the modality-specific offsets in average RT. The Gaussian parameter $\sigma$ displayed a different modality-specific pattern. Neither Gaussian parameters $\mu$ nor $\sigma$ were affected by changes in the event PDF of the input. In contrast, the exponential parameter $\tau$ was sensitive to the event PDF. $\tau$ also captured the difference in RT variance between the two probability distributions in all three sensory modalities. In the long tradition of the ex-Gaussian model of RT the exponential

part has been interpreted as a central and the Gaussian part as a peripheral process[15,35] but also the opposite assignment has been made[15,45,46]. Our purely behavioral study cannot conclusively answer whether the exponential process reflects central, modality-independent computations or whether it reflects more peripheral computations in the neural substrate of each sensory modality. Nonetheless, our behavioral findings suggest that all three sensory modalities share a similar exponential process—which in turn makes specific predictions for what neural activity patterns should be sought on recordings from the relevant sensory and supra-sensory regions.

In sum, we expect that our findings will aid efforts in the understanding of the neural mechanisms involved in predictive processes. The results demonstrate that irrespective of sensory input modality, the brain models its environment's temporal-probabilistic structure using a non-linear transformation of the PDF, but not the hazard rate.

## Methods

**Ethical approval.** The experiments were approved by the Ethics Council of the Max-Planck Society. Written informed consent was given by all participants before the experiment.

**Subjects.** In all, 24 human subjects (13 female), aged 19–33, participated in the auditory, visual, and somatosensory experiments. Of these, 18 subjects (13 female), aged 19–33 participated in the Gaussian control experiment (Supplementary Fig. 14) and 12 other subjects (9 female), aged 20–33 participated in the experiment without catch trials (Supplementary Fig. 13). All were right-handed and had normal or corrected-to-normal vision and reported no hearing impairment and no history of neurological disorder. Participants were naive to the purpose of the experiment. They received €10 per hour for participating.

**Task and procedure.** In auditory, visual, and somatosensory conditions, subjects performed a simple 'set' - 'go' task in which a 'set' cue was followed by a 'go' cue. The timespan between the onset of both cues, termed the 'go' time, was a random variable that was drawn from a specific probability distribution. Subjects were asked to foveate a central black fixation dot and respond as fast as possible to the onset of the 'go' cue with a button press on a response device using the right index finger. After a button press, a small black circle appeared for 0.2 s around the central fixation dot indicating the end of the trial. In some trials, no 'go' cue appeared ('catch trials'), in which case participants were instructed to not press the button. In these catch trials, a small black circle appeared 1.9 s after 'set' cue onset, indicating again the end of the trial. The experiment consisted of two separate sessions taking place at the same time of the day on two consecutive days. A single session consisted of four blocks per sensory modality (vision, hearing, touch) and lasted ~2.5 h. Per block, 165 trials were presented (including 15 'catch trials'), resulting in 1980 trials per session for each subject (3980 trials for two sessions), and a total of 95,040 trials for all subjects. A short training block was run before the first block of each sensory modality on both days to familiarize subjects with the task. During all experimental blocks, subjects wore headphones and positioned their heads on a forehead-and-chin rest (Head Support Tower, SR Research Ltd.) at a fixed distance of 60 cm relative to the computer monitor. Each subject's dominant eye, as determined by Miles test[47] was tracked at a sampling frequency of 1,000 Hz (Eyelink DM-890, SR Research Ltd.). Subjects were asked to restrict eye blinking to the timespan after a button press, i.e. during the ITI. Trials in which visual fixation was not maintained within a radius of 2.5° visual angle around the central fixation point for more than 300 ms during the 'go' time were automatically discarded for data analysis. All stimuli were generated using MatLab (The MathWorks, Natick MA, USA) and the Psychophysics Toolbox[48] on a Fujitsu Celsius M730 computer running Windows 7 (64 bit). The experiment took place in a dimly lit, soundproof booth.

**Auditory stimuli.** Two white noise bursts (50 ms duration, 8 ms cosine ramp, onset and offset) served as 'set' and 'go' cues. They were presented at 60 dB SPL above hearing threshold at 1 kHz, as determined by pure tone audiometry according to ISO 8253. All auditory stimuli were output by a high-quality interface (RME Fireface UCX) and delivered diotically using electrodynamic headphones (Beyer-dynamic DT 770 PRO) driven by a headphone amp (Lake People GT-109). The sound pressure level was calibrated to 75 dB(A) individually for each transducer while using a temporal weighting suited for impulsive stimuli ($l\tau$ = 35 ms). To this end we used an IEC 603184 artificial ear simulator (model G.R.A.S. 43AG) with according pinnae and a IEC 60942 class 1 sound level calibrator (Larson Davis CAL200) and an IEC 942 class 1 pistonphone with barometric correction as calibration source (G.R.A.S. Type 42AA).

**Visual stimuli.** The visual 'set' cue (duration 50 ms) consisted of two checkerboard patterns which were presented simultaneously. One was positioned in an upper quadrant, the other in the lower quadrant on the opposite side. The 'go' cue consisted of two checkerboard patterns the same location but the with black-white pattern reversed. Each checkerboard subtended 6.6 × 6.6 degrees of visual angle and consisted of 7 × 7 black and white squares, each subtending 0.9 × 0.9 degrees of visual angle. The center of each checkerboard was positioned at a horizontal distance of 12.6 degrees of visual angle and at a vertical distance of 7.1° from the center of a central, black fixation dot. The site of presentation alternated between the left and right side on a trial-by-trial basis. Visual stimuli were presented on a BenQ XL2420-B monitor (resolution 1920 × 1080, refresh rate 60 Hz) which was set to a gray background.

**Somatosensory stimuli.** Two short electric pulses (duration 200 µs) were presented as 'set' and 'go' cues using a constant current stimulator (Digitimer DS7A). Each subject's perceptual threshold was determined by increasing stimulus intensity (mean current) until the subject first reported a sensation and then decreasing it until it was no longer perceived. The perceptual threshold was recorded as the lowest ascending stimulus intensity at which the subject reported sensation. For the experimental task, the electric current was set to a higher intensity (mean current = 7.9 ± 3.9 mA) that the subject judged comfortable, yet easily perceptible.

**Temporal probabilities.** The 'go' time was a random variable drawn from one of two probability distributions (Fig. 2b) that was fixed throughout two consecutive blocks of trials. The distributions were chosen by parametrically searching the family of Weibull distributions for cases that would fulfill the two following criteria:

1. The one distribution should be the left-right flipped version of the other so that this symmetry in PDFs would help to identify the effect of elapsed time itself on the modulation of RTs by the probability distributions.
2. The two distributions should have HRs with opposite slopes, in order to investigate the modulation of RTs by HR.

A parametric search identified a Weibull distribution with parameters $k = 1$ and $l = 0.33$:

$$f(t) = \frac{k}{l}\left(\frac{t}{l}\right)^{k-1} e^{-\left(\frac{t}{l}\right)^k} \qquad (1)$$

The identified shape value $k = 1$ reduces the Weibull to an exponential distribution:

$$f(t) = \frac{1}{l} e^{-\left(\frac{t}{l}\right)} \qquad (2)$$

The x-axis of both distributions was discretized with the step size of the monitor refresh rate. This is based on the assumption that this level of discretization is not perceivable in the context of the 'set' - 'go' design, rendering the distributions continuous to the brain. The y-axis was also discretized as the PDF described number of trials at each discrete 'go' time point. Both distributions were delayed by 0.4 s giving a range of 'go' times from 0.4 to 1.4 s. To minimize sequential effects, the 'go' times were randomized with the constraint that no more than two consecutive trials had the same 'go' time. The intertrial interval (ITI, range 1.4 to 2.4 s) was randomly drawn from a uniform distribution. During each session in auditory, visual, and somatosensory conditions half of the blocks followed the exponential distribution (PD$_{Exp}$), the other half followed its flipped counterpart (PD$_{Flip}$). The probabilistic structure changed after two blocks without notification. To control for order effects, the conditions (sensory modalities and probability distributions) were organized in a Latin square design, based on which modality and distribution were shuffled across subjects and days.

**Exponential-Gaussian model.** To quantitatively investigate the distributional properties of RT between conditions, we used the exponential-Gaussian distribution as a well-established parametric two-process model of RT[15,35,45,49].

$$f(t; \mu, \sigma, \tau) = \frac{1}{2\tau} e^{\frac{1}{2\tau}\left(2\mu + \frac{\sigma^2}{\tau} - 2t\right)} erfc\left(\frac{2\mu + \frac{\sigma^2}{\tau} - t}{\sqrt{2}\sigma}\right) \qquad (3)$$

where $erfc$ is the complementary error function defined as

$$erfc(x) = \frac{2}{\sqrt{\pi}} \int_x^\infty e^{-t^2} dt \qquad (4)$$

The parameters $\mu$ and $\sigma$ are the mean and standard deviation respectively of the Gaussian constituent of the ex-Gaussian distribution. Parameter $\tau$ is the exponent of the exponential constituent distribution.

The best-fitting Ex-Gaussian parameters were obtained for each subject in each condition using a least-squares fitting algorithm. We observed no systematic difference in adj. $R^2$ between conditions (Supplementary Fig. 15b). Therefore the ex-Gaussian proved to be a well-fitting model of RT in all conditions.

**Temporally blurred ('subjective') PDFs.** The PDFs were blurred by a temporal uncertainty kernel that scales with elapsed time from a reference time point. More intuitively, the longer the elapsed interval to be estimated, the bigger the uncertainty about its length. In recent work it has been hypothesized that this

uncertainty kernel has a Gaussian shape and its standard deviation increases linearly with time as $\sigma = \varphi \cdot t$, where $t$ is the elapsed time and $\varphi$ is the scale factor by which the standard deviation $\sigma$ of the Gaussian uncertainty kernel increases[5]. This intuitively implies that the ratio of the standard deviation of temporal uncertainty to the elapsed time is constant and for this reason the variable $\varphi$ has been termed a Weber fraction of the estimation of elapsed time[5].

Each of the employed distributions is characterized by the following three functions:

$p(t_{go})$ : Probability Density Function of Reaction Time as a function of 'go' time $t_{go}$

$c(t_{go})$ : Cumulative Density Function of Reaction Time as a function of 'go' time $t_{go}$

$$= \int_0^{t_{go}} p(u)du$$

$h(t_{go})$ : Hazard Rate of Reaction Times as a function of 'go' time $t_{go} = \dfrac{p(t_{go})}{1 - c(t_{go})}$ (6)

Each of the PDFs were blurred with a Gaussian kernel with variance increasing with time. The equations for the corresponding subjective functions are:

$$p_S(t_{go}) = \frac{1}{\varphi t_{go} \sqrt{2\pi}} \int_{-\infty}^{\infty} p(\tau) \cdot e^{-(\tau - t_{go})^2 / (2\varphi^2 t_{go}^2)} d\tau$$ (7)

$$c_S(t_{go}) = \int_0^{t_{go}} p_S(u)du$$ (8)

$$h_S(t_{go}) = \frac{p_S(t_{go})}{1 - c_S(t_{go})}$$ (9)

From Eq. (7) it is evident that for a given 'go' time $t_{go}$ the PDF is convolved with a Gaussian kernel centered at $t_{go}$.

The distributions used in this work were not continuous but discrete. They were only represented at the possible time points of stimulus presentation in either auditory, visual or somatosensory conditions. The stimuli presentation instances for $PD_{Exp}$ and $PD_{Flip}$ ranged between [0.4 1.4] sec at steps of 2/60 s (every 2 frames with frame rate 60 frames per second). The computation of the CDF from the discrete PDF was performed using trapezoidal integration. The hazard rate was computed from these discrete versions of a PDF and a CDF. The relatively small sampling interval resulted in discrete CDFs and hazard rates closely approximating the expected continuous versions of these functions from the analytical solutions (Supplementary Fig. 19).

The shortest 'go' time is 0.4 s after 'set' cue onset. At this time point the uncertainty kernel has standard deviation $\varphi \cdot 0.4$. Similarly at the longest 'go' time of 1.4 s this uncertainty has standard deviation $\varphi \cdot 1.4$. In order to implement Eq. 7 for the computation of the subjective PDF, the definition of the PDF was extended to the left and right of the actual stimulus presentation interval as:

$$p_{padded}(t_{go}) = \begin{cases} 0, (0.4 - 3 \cdot \varphi \cdot 0.4) \le t < 0.4 \\ p(t_{go}), 0.4 \le t \le 1.4 \\ 0, 1.4 > t \ge (1.4 + 3 \cdot \varphi \cdot 1.4) \end{cases}$$ (10)

The extensions were equal to 3 standard deviations of the Gaussian uncertainty function at the shortest and longest 'go' times. This 3-standard-deviation extension was selected as it encapsulates the 99.7% of the Gaussian uncertainty. Then the integral in Eq. (9) was computed between these new extrema $[(0.4 - 3 \cdot \varphi \cdot 0.4), (1.4 + 3 \cdot \varphi \cdot 1.4)]$ instead of the impractical interval of minus to plus infinity. The selection of the value of $\varphi$ was based on previous research[44,50]. With this value of $\varphi = 0.21$ the temporal range of the extended PDF as defined in Eq. (10) becomes [0.148 2.28] s which is also the range of integration in the computation of the subjective PDF in Eq. (7).

The PDF of each distribution, $p(t_{go})$ was normalized so that its integral does not amount to one but to 0.9091, which is the total probability that an event will occur over the timespan covered by the distribution. The remaining 9.09% of trials (30 out of 330) are catch trials in which no 'go' cue was presented. Consequently the maximum value of the resulting CDF $c(t_{go})$ is 0.9091. The catch trials introduced some uncertainty about whether the event will occur at all. The function of the catch trials in the context of modeling temporal-probabilistic structures is to prevent the brain's estimation of a PDF from being confounded by the expectation of a conditional event, i.e. the mandatory occurrence of a 'go' cue at the end of a given 'go' time range. It is easy to see from Eq. (6) that the introduction of catch trials and the corresponding reduction of maximum CDF to a value significantly smaller than one stabilizes the computation of the hazard rate at the right extremum of the PDF, i.e. the part where the CDF would approach one in the absence of catch trials. Here it has to be mentioned that although the maximum CDF value is 0.9091, the denominator in the computation of the HR remain $1-c(t_{go})$ and not $0.9091-c(t_{go})$. This is because this denominator represents the probability that nothing has happened up to time point $t_{go}$. This term includes also the probability that nothing has happened up to $t_{go}$ because the current trial is a catch trial. So this denominator could be alternatively defined as $c_{catch} + c_{max} - c(t_{go})$, where $c_{max}$ is the maximum CDF value of the stimulus PDF, equal to 0.9091,

and $c_{catch}$ is the probability that no 'go' cue appears at all, which is equal to 0.0909. As these two terms sum to one, the denominator in the computation of the HR is correctly stated as in Eq. (6).

**Probabilistically blurred PDFs.** In the definition of the subjective function in Eq. (7), the PDF was convolved with a Gaussian function, which represented the uncertainty in elapsed time estimation at a given time point. This uncertainty has been hypothesized to increase linearly with time[1], irrespective of the probability density function of event occurrence. In addition to this, here an alternative hypothesis was investigated in which the uncertainty in elapsed time estimation depends on the probability density function of event occurrence. The hypothesis states that 'go' times with high probability of event occurrence are associated with low uncertainty in time estimation based on the rationale that the brain predicts the onset of upcoming events of high probability more accurately. In contrast, 'go' times with low probability carry higher uncertainty, even if the time they span is short.

This probabilistic blurring of elapsed time estimation was implemented in a similar fashion to the temporal blurring described in Eq. (7). However the standard deviation of the Gaussian kernel does not scale linearly with time as in the temporal blurring case ($\sigma = \varphi \cdot t$), instead it scales according to the PDF of event occurrence. In order to use realistic variance values in the blurring Gaussian kernel the minimum and maximum values of the standard deviation were set accordingly to the temporal blurring case as

$$\sigma_{min} = \varphi \cdot t_{min} = \varphi \cdot 0.4 \text{ and } \sigma_{max} = \varphi \cdot t_{max} = \varphi \cdot 1.4$$ (11)

The PDF under investigation was then scaled so that its minimum value is $\sigma_{min}$ and its maximum value $\sigma_{max}$.

If $p_{min}$ and $p_{max}$ are the minimum and maximum values respectively of the PDF under investigation then the function used for computing the standard deviation of the Gaussian kernel based on the PDF $p(t)$ was defined as:

$$s(t) = \left[ 1 - \frac{(p(t) - p_{min})}{(p_{max} - p_{min})} \right] \cdot (\sigma_{max} - \sigma_{min}) + \sigma_{min}$$ (12)

From the first term inside the brackets it is obvious that when the probability $p(t)$ is low the standard deviation of the Gaussian kernel approaches $\sigma_{max}$ while when the probability becomes big, $s(t)$ approaches $\sigma_{min}$.

Based on this function for determining the standard deviation of the blurring Gaussian kernel the probabilistically blurred PDF $p_p(t)$ was computed as:

$$p_p(t_{go}) = \frac{1}{s(t)\sqrt{2\pi}} \int_{-\infty}^{\infty} p(\tau) \cdot e^{-(\tau - t_{go})^2 / (2s(t)^2)} d\tau$$ (13)

This equation describes that at a given time instance $t$, according to the PDF of event occurrence, the Gaussian uncertainty on the estimation of elapsed time has standard deviation $s(t)$. The minimum and maximum values of standard deviation $\sigma_{min}$ and $\sigma_{max}$ are set to $\varphi \cdot 0.4$ and $\varphi \cdot 1.4$, as already described earlier. In order to compare this probabilistic blurring hypothesis directly to the initial temporal blurring hypothesis, the value of $\varphi$ was likewise set to 0.21.

Finally, in order to implement the Gaussian blurring of Eq. (13) at the extrema of 'go'-times, the definition of the PDF was extended to the left and right of the actual stimulus presentation interval by three standard deviations of the corresponding smoothing Gaussian kernels, similarly to the temporally blurred case described in Eq. (10), as:

$$p_{padded}(t_{go}) = \begin{cases} 0, (0.4 - 3 \cdot s(0.4)) \le t < 0.4 \\ p(t_{go}), 0.4 \le t \le 1.4 \\ 0, 1.4 > t \ge (1.4 + 3 \cdot s(1.4)) \end{cases}$$ (14)

Notice that the extensions depend on the standard deviation function $s(t)$, which depends on the probability density function.

**Selection of explanatory variables for modeling RT.** The selection of explanatory variables for modeling RT was driven by an expected inverse relation of RT with the PDF and HR of the 'go' times. This is based on the rationale that for high relative values of PDF or HR, RT is expected to be small and vice versa. Previous work demonstrated a negative correlation between the 'subjective' HR and RT in the order of $-0.3$[5]. As mentioned earlier, computation of the HR in the brain would require three separate calculations: computation of PDF, its integration for deriving the CDF, and their division for computing HR. The computation of the PDF is the most basic and necessary step in the sequence outlined above and for this reason it was considered here as an alternative factor that can directly affect RT, even before the CDF and HR are computed.

Here two different functions with an inverse character were selected, a 'mirror' and a 'reciprocal' function. The 'mirror' function just reflects a function mirrored around its mean. Here it is used to capture linear anti-correlations between RT and the explanatory variables PDF and HR.

$$x_{mp}(t_{go}) = -(p(t_{go}) - \bar{p}) + \bar{p} = -p(t_{go}) + 2 \cdot \bar{p}$$ (15)

$$x_{\text{mh}}(t_{\text{go}}) = -(h(t_{\text{go}}) - \bar{h}) + \bar{h} = -h(t_{\text{go}}) + 2 \cdot \bar{h} = -\frac{p(t_{\text{go}})}{1 - c(t_{\text{go}})} + 2 \cdot \bar{h} \quad (16)$$

where

$x_{\text{mp}}$: "mirror"of the PDF
$x_{\text{mh}}$: "mirror"of the hazard rate of the PDF
$\bar{p}$: Mean PDF, $\bar{h}$ : mean HR

$$x_{\text{op}}(t_{\text{go}}) = \frac{1}{p(t_{\text{go}})} \quad (17)$$

$$x_{\text{oh}}(t_{\text{go}}) = \frac{1}{h(t_{\text{go}})} = \frac{1 - c(t_{\text{go}})}{p(t_{\text{go}})} = (1 - c(t_{\text{go}})) \cdot x_{\text{op}}(t_{\text{go}}) \quad (18)$$

where

$x_{\text{op}}$: 'reciprocal' PDF
$x_{\text{oh}}$: 'reciprocal' hazard rate of the PDF.

The 'reciprocal' function simply takes the reciprocal of a function, e.g. 1/PDF. This is used to capture a non-linear anti-correlation.Similar variables are defined for the temporally and probabilistically blurred PDF cases, namely $x_{\text{opt}}$, $x_{\text{oht}}$, $x_{\text{mpt}}$, $x_{\text{mht}}$, for temporal blurring and $x_{\text{opp}}$, $x_{\text{ohp}}$, $x_{\text{mpp}}$, $x_{\text{mhp}}$, where for probabilistic blurring where the third letter in the subscript indicates the type of blurring. As it can be seen in Eq. (18) for any given 'go' time $t_{\text{go}}$, the variable $x_{\text{oh}}(t_{\text{go}})$ is a scaled version of the variable $x_{\text{op}}(t_{\text{go}})$. This scaling depends on the CDF and therefore it is non-linear across 'go' times. The same holds for variables$c(t_{\text{go}})$ $x_{\text{mp}}(t_{\text{go}})$ and $x_{\text{mh}}(t_{\text{go}})$, as can be seen in Eqs. (15) and (16). As the scaling between these pairs of variables is non-linear (dependent on $(1 - c(t_{\text{go}}))$and $\frac{1}{1 - c(t_{\text{go}})}$ respectively) it is also expected that their relation to RT will not be linearly identical. This also justified their treatment as different variables that are not independent but non-linearly related.

**Modeling RT by 'mirror' and 'reciprocal' functions with a linear model**. The eight 'blurred'explanatory variables $x_{\text{opt}}$, $x_{\text{oht}}$, $x_{\text{mpt}}$, $x_{\text{mht}}$, for temporal blurring and $x_{\text{opp}}$, $x_{\text{ohp}}$, $x_{\text{mpp}}$, $x_{\text{mhp}}$ for probabilistic blurring, were constructed to investigate their linear relation to RT. These variables were derived with $\varphi = 0.21$ based on previous research[44,50].

Additionally, the four variables $x_{\text{op}}$, $x_{\text{oh}}$, $x_{\text{mp}}$, and $x_{\text{mh}}$ derived directly from the original PDF, CDF and HR, without any Gaussian blurring, were also fit to RT for comparison. A linear model was built for each of these eight explanatory variables. An Ordinary Least Squares (OLS) regression was employed for the computation of the regression coefficients using the MatLab (The MathWorks, Natick MA, USA) fit function. Any assumption about the distribution of the residuals of the models was omitted, as we had no evidence that they should follow a Gaussian distribution. Adjusted $R^2$ was used as a measure of goodness-of-fit for comparing the different models' relation to RT.

**Comparing temporal and probabilistic blurring**. One of the expected differences between the two blurring methodologies was located at the early and late extrema of the 'go' time interval. This is because in the case of temporal blurring the smoothing kernel has always higher variance at the late extremum, as compared to the early, independent of the PDF used and this should be expected to result in greater smearing of RT curves towards the late extremum due to the always greater uncertainty. This should not be the case in probabilistic blurring, where the standard deviation of uncertainty in interval estimation depends on the PDF used. So for example the uncertainty in the early extremum of the exponential PDF should be very similar to that at the late extremum of the flipped-exponential due to the PDF symmetry. This should also result in identical smearing of the RT curves at these two different extrema for these two different PDFs.

In order to investigate if the explanatory variables based on temporally or probabilistically blurred PDFs capture better the behavior of RT curves at different parts of the 'go' time range, the 'go' time range was divided in 4 equally spaced bins and the goodness-of-fit of all models for the different explanatory variables was computed in each bin. The metric employed for comparing the goodness-of-fit of these models was the sum of squared residuals in each bin.

For each distribution (PD$_{\text{Exp}}$ and PD$_{\text{Flip}}$), each modality (visual, auditory, somatosensory), each of the 4 bins and each of the explanatory variables that was derived based on the blurred PDF, the quotient of the RT model residuals of the temporally blurred case over that of the probabilistically blurred case was computed. This procedure indicated which blurring method provided the model that fits the RT of a specific bin better.

**Modeling of RT relative to auditory condition**. The auditory condition was used as a reference condition and auditory RTs were subtracted from visual RTs and from somatosensory RTs in both PD$_{\text{Exp}}$ and PD$_{\text{Flip}}$ conditions. The resulting across-modality ΔRT curves were fitted with an exponential function of 'go' time:

$$f(t_{\text{go}}) = a \, e^{b \, t_{\text{go}}} + c \quad (19)$$

The condition-specific probability per 'go' time (PD$_{\text{Exp}}$ or PD$_{\text{Flip}}$) was used as weights in the fitting algorithm (ordinary least-squares in MATLAB's fit function,

Fig. 5a). Finally, a combined, additive model of RT was constructed based on the reciprocal, probabilistically-blurred PDF fitted to the auditory RT and the exponential model fitted to the across-modality ΔRT curves:

$$x_{\text{oppc}}(t_{\text{go}}) = \frac{1}{p_p(t_{\text{go}})} + a \, e^{b \, t_{\text{go}}} + c \quad (20)$$

Adjusted $R^2$ was calculated to evaluate the goodness-of-fit of the combined, additive model (Fig. 5b).

**Reporting summary**. Further information on research design is available in the Nature Research Reporting Summary linked to this article.

## Data availability

The data that support the findings of this study are available from the corresponding author upon reasonable request.

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

## Acknowledgements

We thank Caroline Mirkes, Claudia Lehr, Freya Materne, and Cornelius Abel for help with data acquisition and technical support.

## Author contributions

M.G., G.M. and D.P. designed the research. M.G. performed the experiments. M.G. and G.M. analyzed the data. All authors discussed the results. M.G., G.M., L.T.M. and D.P. wrote the paper.

## Competing interests

The authors declare no competing interests.
