## [Peer Review File · Nature Communications]

Reviewers' Comments:

Reviewer #1:

Remarks to the Author:

In this paper, Grabenhorst et al. asked how anticipation of events influences timing behavior. They asked human subjects to make a movement immediately after the presentation of a sensory 'go' cue. The go cue was presented after a delay (tGo), which was a random sample from a fixed distribution; i.e., delay distribution. Following a long tradition in human psychophysics, they evaluated the subjects' ability to anticipate the cue by analyzing the relationship between reaction time (RT) and tGo.

A large body of literature has provided evidence for two hypotheses regarding humans' timing behavior: 1) scalar variability: timing variability scales with elapsed time, and 2) hazard-based anticipation: RTs decrease when the hazard rate of tGo is larger. This paper tests RTs in the presence of two tGo distributions, an exponential distribution, and a flipped exponential distribution, and reports that the results are not consistent with either of these hypotheses. Instead, the paper proposed an alternative model based on two new hypotheses: 1) timing variability depends on the probability distribution of tGo, and not elapsed time, and 2) the anticipatory process is controlled by the probability density function (PDF) of tGo, and not its hazard rate.

The hypotheses are novel and are broadly supported by the results across different sensory modalities. However, given the large body of prior work on this topic, it is really important that the authors make a convincing case. As it stands, the evidence against previous models is weak and unconvincing. More needs to be done to substantiate the claims.

Major comments

The results might not generalize. Evidence against the hazard model was mainly based on data from the flipped distribution. However, results for this condition have to be further scrutinized. First, the PDF-based model with the probabilistic blurring predicts that RTs should be symmetric between the exponential and flipped distributions. However, data indicates that RTs were notably more variable in the flipped condition, which goes against the authors' hypothesis. This is particularly relevant because the hazard rate for the flipped condition exhibits a dramatic ramp, which subjects may not be able to internalize. In other words, the failure of the hazard model in this case may be because subjects weren't able to compute the hazard rate for certain distributions including the flipped exponential, not because the hazard model is wrong. As much as I dislike asking for more experiments, in this case, since the key finding is based on a specific distribution, it is necessary that authors try other distributions to verify the generalizability of their claim more rigorously. For example, they can try a Gaussian distribution for which the two models will have very different predictions. Note that, I realize and appreciate the fact that the results were held across sensory modalities but that is not the right control in this case. The contentious point here is the choice of distribution, and not sensory modality.

Subjects might not have learned the distributions. A related concern is that subjects were confused about the two underlying distributions (and the corresponding hazard functions) because switches between the two were uncued. In other words, their subjective representation of the distribution of 'tGo' could have been different from those imposed experimentally. To address this concern, the authors should either verify independently that the subjective hazards were sufficiently accurate, or at the very least, verify that RT distributions were stable across trials after the switch. A specific relevant analysis is to assess whether RTs long after a switch were better supported by the hazard model (compared to RTs early after the transition).

Current evidence against the hazard model is weak. First, subset of the hazard models (mirror and $1/\text{temp. blur. HR}$ in Fig. 3A and 3C) provided qualitatively correct predictions on subjects' RTs. Indeed, the hazard model and not the PDF-based model was able to capture RT increases for early 'tGo' in the exponential distribution, particularly for auditory and tactile conditions (compare Fig. 3A and 3C to Fig. 4A and 4C). The authors have chosen to ignore this discrepancy, which is odd given that a comparable discrepancy between data and ' $1/\text{temp. blur. PDF}$ ' model (highlighted in Fig. 4B) was used to motivate the probabilistic blurring model. Given that the claims go against long standing ideas, it is critical to verify the weak effects by cross-validation (i.e. splitting data, fitting model for a training set, and measuring fitting performance in the remaining test set).

The transfer functions considered are ad-hoc. To my knowledge, the mirror transfer function has never been seriously proposed, and seems more like a strawman. The one paper that has mentioned something like that is Janssen & Shadlen, 2005, but the reason for that was convenience of comparing neural activity to RT profile without any statement about this representing a transfer function. A more suitable approach for understanding the transfer function may come from the large body of modeling work on RT such as the bounded drift-diffusion models (Ratcliff) or the LATER model (Carpenter). For instance, a potential path forward could be that the anticipation process sets the baseline of a bounded drift-diffusion process. That would lead to a rigorous model for the transfer function.

It is unclear if the results only apply to average behavior across subjects or it explains behavior of individual subjects. Usually, RT varies a lot across subjects and it could have implications on the claim of the paper. It is important to show the behavior and model fits for individual subjects. For instance, is PDF-based model better across all subjects? Is there any difference between subjects experiencing the exponential condition first and ones experiencing later? Along these lines, a problematic aspect of the modeling that has to be remedied is that timing variability is not measured across subjects independently. Instead, a fixed Weber fraction of 0.21 is assumed. At the very least, the authors should fit the Weber fraction as a free parameter or verify that pooling across various Weber fractions does not impact the inferences.

Minor comments

Ref. 9 citation for the transfer function (p. 2) seems wrong.

The normative of the proposed model is not clear. Does the use of PDF have any computational advantage, other than the fact that it described the data better?

Reviewer #2:

Remarks to the Author:

Authors investigated the distribution of reaction time (RT) in a 'set-go' paradigm where the probability of set-go intervals were given as an exponential or flipped-exponential functions with catch trials where go signals were not presented. Those two functions allowed the authors to analyze the contributions of the probability distribution function (PDF) and the hazard rate (HR) to anticipation of sensory events. The authors showed that the PDF based model fitted RT data better than HR based one.

The manuscript is written well. The methods and analysis were well designed. I would just raise one question regarding the clear deviations between the data and the model in sub-second range in PDEXP visual and somatosensory conditions (Fig 4c).

I wonder if the median RT was smaller in the auditory condition than in visual and somatosensory

conditions (modality-specific characteristic) because the distributions of RTs in those two conditions drew U-shapes. In addition, the U-shape distribution in those two conditions could have lead smaller IQRRT in PDEXP than in PDFLIP (modality-independent characteristic). If those suppositions were true, it would not be meaningful to separate modality-specific and -independent characteristics.

I am curious if the authors think that the RTs on the left and right side of the U-shapes (i.e., RTs in the sub and supra-second time range) can be explained by the model presented in this study. Previous studies have suggested that different neural mechanisms are involved in estimation of sub- and supra-second time intervals (for review: Buhushi and Meck, Nat. Rev. Neurosci 2005). I wish the authors add a discussion about how this study is consistent or inconsistent with the previous studies.

Yoshiko Yabe

Reviewer #3:

Remarks to the Author:

This study investigated the effect of the probability that an event will occur on the reaction times, in three sensory modalities: vision, audition and somatosensory. The authors demonstrate that the data can be best fit with the reciprocal of the probability density function in all three modalities, as opposed to the hazard rate of elapsed time.

There are several problems with the manuscript in its present form. First, the authors present two hypotheses about the computations related to the estimation of event occurrence, i.e. the hazard rate and the increasing uncertainty about elapsed time. However, these two hypotheses are not mutually exclusive. The probability distribution itself is blurred because of the scalar property of time perception, which also leads to a blurred hazard rate. It is misleading to present these two mechanisms as two alternative hypotheses.

Secondly, there was no time pressure at all in these experiments, since reward did not depend on reaction time. Therefore, we do not even know to what extent the participants were attending to elapsed time. RT changed over time, but there was no penalty for being slow and no reward for being fast.

I found the distinction between probabilistic and deterministic context confusing. It would be much cleaner to use the PDFs as they are, without truncating any part (i.e. 'deterministic context'). I believe we do not know exactly what the effect is of cutting the tail of the PDF on the participants' perception of the probability of an event. Furthermore, there was no spatial component at all in the task (merely a button press), and this makes the study difficult to compare to previous studies, since the neural representation of elapsed time is frequently reflected in motor planning activity.

Finally and most importantly, the authors make several claims that are not supported at all by the data. This is a purely behavioral study, therefore I cannot see how their findings 'represent a significant contribution towards a mechanistic understanding of long-standing questions' (p. 13). Also, on p. 16 last paragraph the authors state that 'the Gaussian part of the ex-Gaussian model represents modality-specific processes of a peripheral nature, (...), and also shared motor processes'. But there is no evidence at all in the data for this conjecture.

Overall, this study only contributes new data about behavior related to the event probability over time in a very specific context: without time pressure, no spatial component to the response, truncated PDFs, etc.

We thank the reviewers for their thorough and thoughtful comments. The reviewers raised a range of relevant issues that we have addressed with new data and new analyses including those suggested by the reviewers. We reply in detail, interleaving our replies in blue.

The additions to the manuscript support the original conclusions. We recognize as do the reviewers that we are calling into question basic assumptions concerning simple reaction time (RT), specifically that hazard rate (HR) is the mediating variable that controls RT. We believe we make a strong case against HR and we advance an alternative model (1/PDF) that does account for the data.

REVIEWER 1

MAJOR COMMENTS

Reviewer 1: Major Comment 1

Reviewer: The results might not generalize. Evidence against the hazard model was mainly based on data from the flipped distribution. However, results for this condition have to be further scrutinized. First, the PDF-based model with the probabilistic blurring predicts that RTs should be symmetric between the exponential and flipped distributions. However, data indicates that RTs were notably more variable in the flipped condition, which goes against the authors' hypothesis. This is particularly relevant because the hazard rate for the flipped condition exhibits a dramatic ramp, which subjects may not be able to internalize. In other words, the failure of the hazard model in this case may be because subjects weren't able to compute the hazard rate for certain distributions including the flipped exponential, not because the hazard model is wrong.

As much as I dislike asking for more experiments, in this case, since the key finding is based on a specific distribution, it is necessary that authors try other distributions to verify the generalizability of their claim more rigorously. For example, they can try a Gaussian distribution for which the two models will have very different predictions. Note that, I realize and appreciate the fact that the results were held across sensory modalities but that is not the right control in this case. The contentious point here is the choice of distribution, and not sensory modality.

Reply:

The reviewer rightly asked whether our findings generalize to other temporal-probabilistic settings and suggested to repeat the experiment by using another probability distribution, such as the Gaussian, in which the HR- and PDF-based models differ strongly in their predictions. We agree and have done so.

A subset of 18 of the original 24 participants were asked to perform the same set-go experiment in audition, vision and somatosensation with a Gaussian distribution of 'go' times. In this Gaussian condition, the HR-based model failed to capture the RT curves even in a *qualitative* way (**Fig. 5a** in updated manuscript). Although the RT curves followed a U-shape, the HR-based model predicted a monotonically decreasing one. In contrast, the PDF-based model qualitatively captured the data in all three sensory modalities. The fit was

exceptionally accurate in audition (**Fig. 5b** in updated manuscript). These results are in agreement with those previously reported in the manuscript regarding the exponential and flipped conditions - and very much support the hypothesis that the brain uses the PDF and not the HR to model event probability across time.

We also note that the discrepancies between HR-based model and data in our experiment in the flipped exponential condition are of a qualitative nature (concavity in RT curves vs. convexity in HR-based model), i.e. the HR-based model makes inadequate predictions even at shorter 'go' times where there is no steep increase in HR.

A new section on the Gaussian condition, including **Fig.5**, has been added to the Results section of the manuscript on p. 11-12.

Reviewer 1: Major Comment 2

Reviewer: Subjects might not have learned the distributions. A related concern is that subjects were confused about the two underlying distributions (and the corresponding hazard functions) because switches between the two were uncued. In other words, their subjective representation of the distribution of 'tGo' could have been different from those imposed experimentally.

To address this concern, the authors should either verify independently that the subjective hazards were sufficiently accurate, or at the very least, verify that RT distributions were stable across trials after the switch. A specific relevant analysis is to assess whether RTs long after a switch were better supported by the hazard model (compared to RTs early after the transition).

Reply:

Agreed. As suggested, we performed a split-block analysis on the data in the exponential and flipped exponential conditions. Each of the six experimental conditions (exponential and flipped exponential PDF in audition, vision and somatosensation) consisted of four blocks of trials. We selected the pair of blocks performed on day #1 and another pair performed on day #2. The distribution was constant within each pair and changed after each pair. We selected the first 100 trials of the first block of a pair to represent behavior immediately after a switch in distribution (early period), and we selected the last 100 trials of the second block of each pair to represent behavior just before a switch in distribution (late period), i.e. when participants should have learned the distributions. We performed the same modeling procedure with the HR-based and the PDF-based model as we did on the complete data set. The results are presented in Suppl. Figs. 4 and 5.

RT curves were consistent across all early and late periods in each distribution-modality condition. In both early and late periods, the modeling confirmed what was seen in the modeling of the complete data: in the exponential condition, the HR models qualitatively captured the data while in the flipped condition they failed to do so, even in the late period. In contrast, in both exponential and flipped exponential conditions, the PDF-based models provided a good fit to the data in both early and late periods. These results provide evidence that even after many trials of the same distribution, when participants are more likely to

have learned its temporal-probabilistic structure, the HR *still* fails to account for the RT curves, while the PDF does.

We added this analysis to the Results section on p. 9 and added Suppl. Figs. 4 and 5.

Reviewer 1: Major Comment 3

Reviewer: Current evidence against the hazard model is weak.

First, subset of the hazard models (mirror and 1/ temp. blur. HR in Fig. 3A and 3C) provided qualitatively correct predictions on subjects' RTs. Indeed, the hazard model and not the PDF-based model was able to capture RT increases for early 'tGo' in the exponential distribution, particularly for auditory and tactile conditions (compare Fig. 3A and 3C to Fig. 4A and 4C).

The authors have chosen to ignore this discrepancy, which is odd given that a comparable discrepancy between data and '1/ temp. blur. PDF' model (highlighted in Fig. 4B) was used to motivate the probabilistic blurring model.

Given that the claims go against long standing ideas, it is critical to verify the weak effects by cross validation (i.e. splitting data, fitting model for a training set, and measuring fitting performance in the remaining test set).

Reply:

The reviewer recommended a specific analysis based on cross-validation to investigate the strength of the effects:

"Given that the claims go against long standing ideas, it is critical to verify the weak effects by cross validation (i.e. splitting data, fitting model for a training set, and measuring fitting performance in the remaining test set)."

We followed this helpful recommendation by fitting both the HR-based and the PDF-based model to RT from the first half of the experiment and examined how these fitted models relate to the data from the later half of the experiment. We found that both key findings hold in this analysis: the PDF-based model outperforms the HR-based model and the probabilistic blurring adequately fits the RT at longer 'go' times. We added a section in the Results (p. 9) and a figure in the supplement based on this analysis (Supplementary Fig. 6).

The upward bend: We agree that there are upward bends in RT curves during short 'go' times in vision and somatosensation, evident in the case of the exponential PDF. Indeed during this early 'go' time period there is a significant deviation between RTs and the PDF-based model. However, the HR-based model, although it appears to qualitatively be more suitable as it demonstrates a pronounced upward bend, actually also has large deviations from the RTs, larger than the PDF-based model.

In all sensory modalities, the HR-model's predictions are more than 50 ms larger than RT at the shortest GT, which is problematic because these are the timepoints of highest event probability in the exponential condition. In the case of the PDF-based models (Fig. 4 c), the deviations are smaller – the model underestimates the data by 20 - 25 ms in vision and somatosensation and much less in audition.

Qualitatively, both the HR-based and the PDF-based models feature an upwards bend which is caused by the blurring with a Gaussian kernel representing uncertainty in time estimation. In the HR-based model, this upwards bend is over-represented, as evidenced by the large differences between data and model mentioned above. In the PDF-based model, the upwards bend is under-represented (one needs to really zoom into the figure to notice it).

Qualitatively both models have the potential to capture the early bend in RT curves. However it should be seriously considered whether these early bends represent a component of the modeling of event probability by the brain or whether they are correlates of modality-specific processing at short time-scales. It is obvious that in the case of audition the upward bend in RT in the exponential condition is almost absent. This indicates that in the auditory modality the brain manages to anticipate well the instance of highest event probability, even if this instance is at very short 'go' times. It is in the other 2 modalities, vision and somatosensation, in which the upward bend is more pronounced and it might be related to the processing of events occurring at such short 'go' times and not with the probability distribution itself. If this were the case, then this increase in RT towards shorter 'go' times would be visible irrespective of the probability distribution of events. In the case of the flipped exponential, the bend in the RT at shorter 'go' times is not observable directly because already the longest RTs are at the shortest 'go' times. But a closer look shows that the RTs in vision and somatosensation at the shorter 'go' times are much longer than in audition.

Fig. R1 depicts the difference between visual and auditory RT (green curve) and somatosensory and auditory RT (orange curve). It is obvious that both vision and somatosensation feature longer RTs than audition, and this difference in RT increases towards shorter 'go' times. Most importantly this seems to occur regardless of the type of event probability distribution. The reduced effect of this early-'go'-time-phenomenon in the auditory modality can be also clearly seen in the Gaussian condition (Fig. 5 in updated manuscript). There a similar early upward bending of RTs is evident in the other 2 modalities but not in audition (Fig. 5b).

Fig. R1

In contrast to the above modality-specific findings for short go-times, the improvement in model fit due to probabilistic blurring at late 'go' times is not modality-specific. In Fig. 4b the deviations between data and model at the longer 'go' times are very similar in magnitude

across the three modalities. Note that at shorter 'go' times there are also deviations between data and model, e.g. in all three modalities the model under-estimates RT at around 0.6 to 0.8 s. The probabilistically blurred model improves the model fit in these aspects (Fig. 4d). These findings generalize across the three modalities as can be seen in the residuals plot comparing the temporally- and probabilistically blurred models (Fig. 4f).

We have added a new section in the results (pp. 12 and 13) and a new Fig. (Fig. 6) describing the modality-specific up-ward bends at shorter 'go' times and how this might represent processes not related to event probability estimation and discuss (pp. 19 and 20) the possible mechanisms responsible for it.

Reviewer 1: Major Comment 4

The transfer functions considered are ad-hoc. To my knowledge, the mirror transfer function has never been seriously proposed, and seems more like a strawman. The one paper that has mentioned something like that is Janssen & Shadlen, 2005, but the reason for that was convenience of comparing neural activity to RT profile without any statement about this representing a transfer function. A more suitable approach for understanding the transfer function may come from the large body of modeling work on RT such as the bounded drift-diffusion models (Ratcliff) or the LATER model (Carpenter). For instance, a potential path forward could be that the anticipation process sets the baseline of a bounded drift-diffusion process. That would lead to a rigorous model for the transfer function.

Thank you for raising this valid concern about the potentially misleading terminology regarding the use of "transfer function". We use this term to describe the relation between the output (RTs) and the inputs (PDF or HR) of a participant's 'behavioral black box' (brain). In this sense our analysis is indeed similar to that of Janssen & Shadlen, 2005, relating neural activity to HR and relating RT to HR, both by anti-correlation, which is a linear transformation. It is also similar to Schoffelen et al, where RT and HR were related through linear anti-correlation. In the same work, also beta and gamma power were linearly related to HR. Similarly, Ghose and Maunsell (Nature, 2002) linearly relate spike rates to HR by linear correlation and regression. Also Nobre & van Ede in their review about temporal structure and attention (Nat. Rev. Neurosci, 2018) highlight the importance of the HR as a prominent model in predictive behavior, and they cite the above literature as state-of-the-art knowledge. Therefore, a linear relationship between HR and behavior or between HR and neural activity is a very common hypothesis – and arguably the dominant view. It is this linear transformation that we refer to when we use the term "transfer function" in our manuscript. Of course this type of "transfer function" is very different from what is typically described by this term in the literature on bounded drift diffusion models (DDM). To remove any confusion for the reader regarding the term "transfer function", we substituted it in the manuscript by the term "transformation" (p. 3). Fig. 1b was changed accordingly.

DDMs are mathematical models describing behavior and neural processing on Marr level 3 (algorithmic implementation). Here, we are focussed on a clear behavioral hypothesis: what function of the event PDF controls RT? We show that – *whatever* the neural processing on Marr level 3 – HR is not the appropriate model of human performance on Marr level 2. Our

results constrain all models including those derived in a DDM framework. We agree that it would be extremely interesting and valuable to examine how our results could or could not be modeled in any particular DDM framework, but this would take us far from the focus of our paper.

Reviewer 1: Major Comment 5

It is unclear if the results only apply to average behavior across subjects or it explains behavior of individual subjects. Usually, RT varies a lot across subjects and it could have implications on the claim of the paper. It is important to show the behavior and model fits for individual subjects. For instance, is PDF-based model better across all subjects? Is there any difference between subjects experiencing the exponential condition first and ones experiencing later?

The reviewer, reasonably, stresses the importance of single-subject modeling to support the manuscript's claims. To address this significant concern, we fitted both the HR-based and the PDF-based models to RT from single subjects. Four new supplemental figures were added for this single-subject analysis. Suppl. Fig. 10 presents the adjusted R^2 of the single-subject fits for the exponential and the flipped exponential distributions. It is very clear that the PDF-based model outperformed the HR-based one. In Suppl. Fig. 11, the scatter plots depict individual subjects' adjusted R^2 for the PDF-based model versus the HR-based model. It is obvious that the vast majority of subjects' data points were located over the diagonal, indicating that the PDF-based model in the majority of subjects performed better than the HR-based model in all modalities and event distributions. To give representative examples of the single subject data and model fits of both HR-based and PDF-based models, two figures show the analysis of individual subjects: Suppl. Fig. 8 depicts RT curves for two different subjects with high adjusted R^2 and Suppl. Fig. 9 shows fits from two different subjects' with low adjusted R^2 . Even in the low adjusted R^2 cases, it is evident that the key findings remain the same: the PDF-based model qualitatively captures the data better than the HR-based model.

We added a part on this analysis to the Results section (p. 10) and the four figures to the Supplement (Suppl. Figs. 8-11).

The reviewer further suggested to investigate the impact of the sequence of the presented event PDF:

"Is there any difference between subjects experiencing the exponential condition first and ones experiencing later? "

We addressed this question with a specific new analysis. For half of the participants, the sequence of event PDFs was $PD_{Exp} - PD_{Flip}$, for the other half the sequence was $PD_{Flip} - PD_{Exp}$. For each of the two groups mean RT was calculated per 'go' time (Suppl. Fig. 7). We found that RT curves were highly similar in both groups of subjects indicating no strong effect of the sequence of distributions.

A figure was added to the Supplement (Suppl. Fig. 7) and a part was added to the Results (pp. 9-10).

Reviewer 1: Major Comment 6

Along these lines, a problematic aspect of the modeling that has to be remedied is that timing variability is not measured across subjects independently. Instead, a fixed Weber fraction of 0.21 is assumed. At the very least, the authors should fit the Weber fraction as a free parameter or verify that pooling across various Weber fractions does not impact the inferences.

The reviewer suggests, plausibly, that each participant has individual timing variability, represented by a subject-specific Weber fraction, Φ . The pooling across these individual Weber fractions might impact the modeling results and the corresponding inferences. We thank the reviewer for bringing this to our attention. We carried out the analyses suggested by the reviewer.

In interval timing tasks, a Gaussian kernel with the Weber fraction, Φ as its standard deviation is commonly used to account for the uncertainty in time estimation. The value of $\Phi = 0.21$ that was used in our work has been shown in the literature to represent the average timing variability in humans (Buetti et al. J Neurosci 2010). Similar values for Φ are also used in work on non-human primates, e.g. 0.26 (Janssen and Shadlen, 2005). For this reason we did not expect significant differences in the average model fit when individual values of Φ were used.

However, we wanted to explicitly address the reviewer's suggestion that for each subject an individual Weber fraction should be used. Based on the literature (Mauk and Buonomano, 2004), we selected a range of Φ values between 0.1 and 0.3 in steps of 0.01 and fitted both the HR-based and the PDF-based models to individual subject's RT data from each experimental condition (event distribution - sensory modality). For each subject, we selected a subject-specific Φ value, which was the one that had the smallest variance in adjusted R^2 across the six experimental conditions, i.e. the Φ that gave the most stable adjusted R^2 across all conditions for the same subject. This subject-specific Φ value was then used to fit the models again to the six conditions in each individual subject. Then for each experimental condition, the resulting models for the 24 participants were averaged, at each 'go' time. This average model across subjects was then plotted over the average RT across subjects and the adjusted R^2 was calculated. As can be seen Suppl Fig. 12, the model fits are very similar to the ones reported earlier in the manuscript (Figs. 3a, 3b, 4c, 4d) confirming that the initially selected value of $\Phi = 0.21$ was a good approximation of average timing uncertainty. The above analysis yielded an average value of $\Phi = 0.2054$ for the HR-based models and $\Phi = 0.1846$ for the PDF-based models, which were quite close to the initial value of $\Phi = 0.21$ selected for the group-level analyses.

We added a figure on this important control analysis to the supplementary material (Suppl. Fig. 12), addressed the analysis in the Results section of the manuscript (pp. 10-11), and added Buetti et al. J Neurosci, 2010 as a reference.

Reviewer 1: MINOR COMMENTS

Reviewer 1: Minor Comment 1

Ref. 9 citation for the transfer function (p. 2) seems wrong.

We apologize for the lack of clarity regarding the terminology and the citation.

As we mentioned above in our reply to comment 4, we have changed the terminology.

In the cited paper, Schoffelen et. al. Science, 2005, the authors show a clear anti-correlation between RT and HR model (Figs. 2a and 3a). The authors strongly argue for this linear anti-correlated relation between RT and Hazard Rate. In other words, while the authors do not explicitly use the term *transfer function* in their paper – and we do not claim they do – their analysis clearly *implies* a linear transfer function between RT and HR.

Reviewer 1: Minor Comment 2

The normative of the proposed model is not clear. Does the use of PDF have any computational advantage, other than the fact that it described the data better?

The normative value of the PDF model we propose, and its computational advantage, is best appreciated when directly compared to the HR model. The computation of HR is notoriously unreliable (Luce, *Response times*, 1986 p. 123; Rice & Rosenblatt, *The Indian Journal of Statistics*, 1976). Even small errors in the representation of PDF or its CDF will lead to large and unpredictable errors in HR, which could have considerable consequences for an organism relying on the HR as its model of temporal probability. The term 1 / PDF, on the other hand, can be interpreted as "one-in-many" – a simpler and more stable computation. For example, if the probability of an event is 0.02, the brain might interpret it as "1/0.02", i.e. "one-in-fifty".

Another aspect that makes the reciprocal PDF a more plausible candidate employed by the brain for learning probabilities is its close resemblance to the "surprisal" in the context of Shannon's information theory. "Surprisal" is defined as as the logarithm of the reciprocal of outcome probability, namely $\log_2(1/p(x))$, where $p(x)$ is the probability of the outcome x . It is clear that the reciprocal pdf of an event is closely related and can actually be transformed into the outcome probability $p(x)$, with a sum over the entire range of possible outcomes equal to 1. Our results indicate that the brain might actually be computing a simplified form of Shannon information.

To further clarify our hypotheses regarding the choice of the PDF-based model and its possible computational implications for the underlying brain mechanisms, we added a brief part to the Introduction (p. 4) and to the Discussion (pp. 18-19).

REVIEWER 2

Reviewer 2: Comment 1

I would just raise one question regarding the clear deviations between the data and the model in sub-second range in PDEXP visual and somatosensory conditions (Fig 4c). I wonder if the median RT was smaller in the auditory condition than in visual and somatosensory conditions (modality-specific characteristic) because the distributions of RTs in those two conditions drew U-shapes. In addition, the U-shape distribution in those two conditions could have lead smaller IQRRT in PDEXP than in PDFLIP (modality-independent characteristic). If those suppositions were true, it would not be meaningful to separate modality-specific and -independent characteristics.

This is a very valid concern raised by the reviewer, and it is indeed a very good question whether the upward bends in RTs of the visual and somatosensory modalities for the exponential distribution are driving the main differences observed in median RT between modalities and in variability of RT within modalities.

To address this plausible concern, we repeated the analyses after removing all RTs from the short 'go' times constituting the up-wards bend, i.e. only RTs from the 'go' times range of 0.5667 s to 1.4 s were used. This selection reduced the size of the dataset in the exp condition more than in the flip condition, because in exp the probability of short GT is much higher than in flip. Nonetheless, the patterns observed in the full dataset (Figs. 7b and 7c) were also seen in these truncated datasets: i) median RT was significantly shorter in the auditory condition when compared to vision and somatosensation, ii) within modality, median RT did not differ between exponential and flipped exponential distributions, iii) the interquartile range of RT differed across exponential and flipped exponential distributions in vision and somatosensation (Suppl. Fig. 15). These results are in full agreement with those obtained from the whole dataset and constitute evidence against the hypothesis that the early upward bends in RT curves were the driving factors behind these effects. The only difference that was found in the truncated dataset was in audition: IRT RT was larger in the flipped exponential than in the exponential distribution, the effect did not reach significance in audition ($P=0.27$). This could not have been caused by an upwards bend because such a prominent bend does not exist in the auditory condition. Instead, it is probably caused by the data selection which removed a lot of data points.

We addressed the above control analysis in the Results section of the manuscript on p. 15 and added Suppl. Fig. 15 to the supplement.

Reviewer 2: Comment 2

I am curious if the authors think that the RTs on the left and right side of the U-shapes (i.e., RTs in the sub and supra-second time range) can be explained by the model presented in this study. Previous studies have suggested that different neural mechanisms are involved in estimation of sub- and supra-second time intervals (for review: Buhushi and Meck, Nat. Rev.

Neurosci 2005). I wish the authors add a discussion about how this study is consistent or inconsistent with the previous studies.

The reviewer rightly asks whether the PDF-based model we propose in our manuscript can explain the RTs in both the sub- and supra-second range. To address this important aspect, we added a new analysis and figure (Fig. 6 in the updated manuscript) to the Results section (p. 12) suggesting a way in which the PDF-based model may account for the modality-specificity in RT at shorter 'go'-times, e.g. the U-shape.

The reviewer's comment raises another very interesting question: how do the potentially different mechanisms that have been suggested in the literature to underly timing behavior at different timescales relate to the RT modulation observed in our manuscript? We find this question to be very relevant and have added a part to the Discussion (pp. 19-20) where we relate our results to the literature on prominent neural mechanisms commonly suggested to be involved in the estimation of sub- and supra-second time intervals.

REVIEWER 3

Reviewer 3: Comment 1

There are several problems with the manuscript in its present form. First, the authors present two hypotheses about the computations related to the estimation of event occurrence, i.e. the hazard rate and the increasing uncertainty about elapsed time. However, these two hypotheses are not mutually exclusive. The probability distribution itself is blurred because of the scalar property of time perception, which also leads to a blurred hazard rate. It is misleading to present these two mechanisms as two alternative hypotheses.

We apologize for a lack of clarity in our presentation. We agree, of course, that the hazard rate model and the uncertainty in elapsed time estimation are not mutually exclusive, and we did not intend to present them as a two *competing* hypotheses. Instead, what we tried to convey - not very successfully due to our failure to be sufficiently clear - was that there are two prominent sources of uncertainty involved in temporal anticipation: (i) the probability distribution of events in time and (ii) the uncertainty in the estimation of elapsed time. Importantly, for each of these sources of uncertainty, a canonical hypothesis has been provided by previous work, and both hypotheses combine to describe the brain's predictive efforts.

In our manuscript, we use the concept of *mapping rules* to describe the combination of the two hypotheses. The HR (*Hypothesis A*) and the scalar property in time estimation (*Hypothesis B*) together constitute such a mapping rule that is exactly the blurred hazard rate that the reviewer mentions. Our analysis evaluated different mapping rules to identify the one which could explain best the RT curves.

To address the reviewer's concerns, we now clearly state in Fig. 1B that a "Mapping rule" comprises both Hypotheses A and B, and we changed the corresponding section in the Introduction on p. 3. We now also state clearly that hypotheses A and B are neither mutually exclusive (p. 2, line 3) nor are they considered as independent and added a part of the reviewer's statement about the blurred hazard rate (p. 2, beginning of last paragraph).

Reviewer 3: Comment 2

Secondly, there was no time pressure at all in these experiments, since reward did not depend on reaction time. Therefore, we do not even know to what extent the participants were attending to elapsed time. RT changed over time, but there was no penalty for being slow and no reward for being fast.

The reviewer rightly raises a very important aspect of these (types of) experiments: the influence of time pressure on reaction time by the introduction of reward for fast responses. While reward is a driving force on behavior in a multitude of value-based tasks, it is important to note that we deliberately did not include this experimental manipulation into our experiments. **The reason for this is that almost all previous work in the area of temporal-probabilistic inference in humans does not impose explicit loss functions such as "time pressure"**. In order to be able to compare our work to earlier work, we avoided adding such explicit loss functions in this experiment. Certainly, the advantages of using an explicit loss function are many. In particular, we can derive optimal strategies maximizing expected gain and compare human performance to a normative ideal. We are, indeed, developing a Bayesian decision-theoretic model along these lines. Such a model, however, will not allow us to test previous claims based on experiments without explicit loss functions as we do here.

Reviewer 3: Comment 3

I found the distinction between probabilistic and deterministic context confusing. It would be much cleaner to use the PDFs as they are, without truncating any part (i.e. 'deterministic context'). I believe we do not know exactly what the effect is of cutting the tail of the PDF on the participants' perception of the probability of an event.

We apologize for our lack of clarity. We did not truncate the distributions differently between the conditions. The PDF form remained exactly the same, but the density values were normalized so that the integral reflected the percentage of catch trials. The integral was 1 for 0 % catch trials (deterministic condition) and 0.9091 for 9.09 % catch trials (probabilistic condition).

We used the terms "probabilistic" and "deterministic" as they are used in a part of the anticipation literature (e.g. Starkweather et al., Nat Neurosci 2017) to describe a state of event certainty (deterministic state = no catch trials) and a state of event uncertainty (probabilistic state = catch trials possible).

We thank the reviewer for pointing out this potential source of confusion and conclude that our use of the terms "probabilistic" and "deterministic" was evidently misleading. We changed the respective parts in our manuscript by substituting the term "probabilistic" to "context with catch trials" and substituted the term "deterministic" by "context without catch trials" on p. 10 and on pp. 18 and 19.

Reviewer 3: Comment 4

Furthermore, there was no spatial component at all in the task (merely a button press), and this makes the study difficult to compare to previous studies, since the neural representation of elapsed time is frequently reflected in motor planning activity.

A button press is an action or motor response and we expect there would be a neural representation of elapsed time.

However, to investigate the HR hypothesis, we deliberately studied simple RTs. If there were a choice between response locations, we would be studying choice reaction times, and our resulting models would include a necessarily speculative model of the "choice" process. We would no longer be able to directly compare our results to the large literature on simple RT and HR.

Reviewer 3: Comment 5

Finally and most importantly, the authors make several claims that are not supported at all by the data. This is a purely behavioral study, therefore I cannot see how their findings 'represent a significant contribution towards a mechanistic understanding of long-standing questions' (p. 13). Also, on p. 16 last paragraph the authors state that 'the Gaussian part of the ex-Gaussian model represents modality-specific processes of a peripheral nature, (...), and also shared motor processes'. But there is no evidence at all in the data for this conjecture.

Agreed. To address the reviewer's concerns, we have removed the first statement (p. 13) and rewrote a section in the Discussion (p. 21 in updated manuscript).

We did not intend the sentence on p. 13 to be a claim about neural structure (Marr Level 3) but to be a claim about components ("mechanisms") of a behavioral model (Marr Level 2). The term "mechanism" is commonly used in such models (e.g. "opponent-color mechanisms"). We have decided to remove the sentence entirely to avoid any possible confusion.

We have removed the second claim (p.16) and instead on p. 21 now clearly state that our behavioral study cannot answer whether the exponential component reflects central or peripheral computations. We conclude that the modality-independence of the exponential component, as seen in behavior, may guide the identification of computations related to anticipation in the analysis of central and peripheral neural activity patterns.

Reviewer 3: Comment 6

Overall, this study only contributes new data about behavior related to the event probability over time in a very specific context: without time pressure, no spatial component to the response, truncated PDFs, etc.

What the reviewer sees as a weakness of our study, we interpret as a strength. We aimed to have a very well-defined and restricted experiment so that the interpretation we offer is precise and compelling.

It is customary in discriminating between models to pick the simplest possible experiment and to design experiments in light of previous, similar work. The focus of our experiments was on testing the HR hypothesis, and we show there is strong evidence against it -- across three sensory modalities. We propose a novel model of simple RT based on a large amount of data. We provide considerable evidence against a commonly accepted model of simple RT based on HR. Our results are directly comparable to previous work in this field where there is typically no time pressure, etc. We deliberately avoided imposing loss functions (including "time pressure") in this work to maintain comparability. We also considered only simple RT not choice RT with multiple possible responses varying across space.

We agree that each of the potential directions for future research outlined by the reviewer is very reasonable and important.

It will be very exciting to extend this work to account for choice RT with multiple responses distinguished in space where we add a potentially complex decision process to the factors we evaluate.

We are also evaluating the effect of imposing loss functions in time (within the framework of Bayesian decision theory) on the distribution of RT. We have considerable past experience in doing so:

Dean, M., Wu, S.-W. & Maloney, L. T. (2007), Trading off speed and accuracy in rapid, goal directed movements, *Journal of Vision*, 7(5):10, 1-12.;

Hudson, T. E., Maloney, L. T. & Landy, M. S. (2008), Optimal compensation for temporal uncertainty in movement planning. *PLoS Computational Biology*, 4(7):e100130, 1-9.

Wu, S.-W., Dal Martello, M. F. & Maloney, L. T. (2009), Sub-optimal tradeoff of time in sequential movements. *PLoS ONE*, 4(12): e8228, 1-13.

Zhang, H., Wu, S.-W. & Maloney, L. T. (2010), Planning multiple movements within a fixed time limit: The cost of active time allocation in a visuo-motor task. *Journal of Vision*, 10(6):1, 1-17.

*Trommershäuser, J., Maloney, L. T. & Landy, M. S. (2008), Decision making, movement planning and statistical decision theory. *Trends in Cognitive Science*, 12(8), 291-297. [Review]

*Maloney, L. T. & Zhang, H. (2010), Decision-theoretic models of visual perception and action. *Vision Research*, 50, 2362-2374. [Review]

As we discuss on p. 17, based on our experiment we can now design loss functions and choose PDFs that allow further rigorous comparison between different models of RT in a Bayesian decision theoretic framework. This work will necessarily require many new experiments.

Original Reviews

Reviewer #1 (Remarks to the Author):

In this paper, Grabenhorst et al. asked how anticipation of events influences timing behavior. They asked human subjects to make a movement immediately after the presentation of a sensory 'go' cue. The go cue was presented after a delay (tGo), which was a random sample from a fixed distribution; i.e., delay distribution. Following a long tradition in human psychophysics, they evaluated the subjects' ability to anticipate the cue by analyzing the relationship between reaction time (RT) and tGo.

A large body of literature has provided evidence for two hypotheses regarding humans' timing behavior: 1) scalar variability: timing variability scales with elapsed time, and 2) hazard-based anticipation: RTs decrease when the hazard rate of tGo is larger. This paper tests RTs in the presence of two tGo distributions, an exponential distribution, and a flipped exponential distribution, and reports that the results are not consistent with either of these hypotheses. Instead, the paper proposed an alternative model based on two new hypotheses: 1) timing variability depends on the probability distribution of tGo, and not elapsed time, and 2) the anticipatory process is controlled by the probability density function (PDF) of tGo, and not its hazard rate.

The hypotheses are novel and are broadly supported by the results across different sensory modalities. However, given the large body of prior work on this topic, it is really important that the authors make a convincing case. As it stands, the evidence against previous models is weak and unconvincing. More needs to be done to substantiate the claims.

Major comments

The results might not generalize. Evidence against the hazard model was mainly based on data from the flipped distribution. However, results for this condition have to be further scrutinized. First, the PDF-based model with the probabilistic blurring predicts that RTs should be symmetric between the exponential and flipped distributions. However, data indicates that RTs were notably more variable in the flipped condition, which goes against the authors' hypothesis. This is particularly relevant because the hazard rate for the flipped condition exhibits a dramatic ramp, which subjects may not be able to internalize. In other words, the failure of the hazard model in this case may be because subjects weren't able to compute the hazard rate for certain distributions including the flipped exponential, not because the hazard model is wrong. As much as I dislike asking for more experiments, in this case, since the key finding is based on a specific distribution, it is necessary that authors try other distributions to verify the generalizability of their claim more rigorously. For example, they can try a Gaussian distribution for which the two models will have very different predictions. Note that, I realize and appreciate the fact that the results were held across sensory modalities but that is not the right control in this case. The contentious point here is the choice of distribution, and not sensory modality.

Subjects might not have learned the distributions. A related concern is that subjects were confused about the two underlying distributions (and the corresponding hazard functions) because switches between the two were uncued. In other words, their subjective representation of the distribution of 'tGo' could have been different from those imposed experimentally. To address this concern, the

authors should either verify independently that the subjective hazards were sufficiently accurate, or at the very least, verify that RT distributions were stable across trials after the switch. A specific relevant analysis is to assess whether RTs long after a switch were better supported by the hazard model (compared to RTs early after the transition).

Current evidence against the hazard model is weak. First, subset of the hazard models (mirror and $1/\text{temp. blur. HR}$ in Fig. 3A and 3C) provided qualitatively correct predictions on subjects' RTs. Indeed, the hazard model and not the PDF-based model was able to capture RT increases for early 'tGo' in the exponential distribution, particularly for auditory and tactile conditions (compare Fig. 3A and 3C to Fig. 4A and 4C). The authors have chosen to ignore this discrepancy, which is odd given that a comparable discrepancy between data and ' $1/\text{temp. blur. PDF}$ ' model (highlighted in Fig. 4B) was used to motivate the probabilistic blurring model. Given that the claims go against long standing ideas, it is critical to verify the weak effects by cross-validation (i.e. splitting data, fitting model for a training set, and measuring fitting performance in the remaining test set).

The transfer functions considered are ad-hoc. To my knowledge, the mirror transfer function has never been seriously proposed, and seems more like a strawman. The one paper that has mentioned something like that is Janssen & Shadlen, 2005, but the reason for that was convenience of comparing neural activity to RT profile without any statement about this representing a transfer function. A more suitable approach for understanding the transfer function may come from the large body of modeling work on RT such as the bounded drift-diffusion models (Ratcliff) or the LATER model (Carpenter). For instance, a potential path forward could be that the anticipation process sets the baseline of a bounded drift-diffusion process. That would lead to a rigorous model for the transfer function.

It is unclear if the results only apply to average behavior across subjects or it explains behavior of individual subjects. Usually, RT varies a lot across subjects and it could have implications on the claim of the paper. It is important to show the behavior and model fits for individual subjects. For instance, is PDF-based model better across all subjects? Is there any difference between subjects experiencing the exponential condition first and ones experiencing later?

Along these lines, a problematic aspect of the modeling that has to be remedied is that timing variability is not measured across subjects independently. Instead, a fixed Weber fraction of 0.21 is assumed. At the very least, the authors should fit the Weber fraction as a free parameter or verify that pooling across various Weber fractions does not impact the inferences.

Minor comments

Ref. 9 citation for the transfer function (p. 2) seems wrong.

The normative of the proposed model is not clear. Does the use of PDF have any computational advantage, other than the fact that it described the data better?

Reviewer #2 (Remarks to the Author):

Authors investigated the distribution of reaction time (RT) in a 'set-go' paradigm where the probability of set-go intervals were given as an exponential or flipped-exponential functions with catch trials where go signals were not presented. Those two functions allowed the authors to analyze the contributions of the probability distribution function (PDF) and the hazard rate (HR) to anticipation of sensory events. The authors showed that the PDF based model fitted RT data better than HR based one.

The manuscript is written well. The methods and analysis were well designed. I would just raise one question regarding the clear deviations between the data and the model in sub-second range in PDEXP visual and somatosensory conditions (Fig 4c).

I wonder if the median RT was smaller in the auditory condition than in visual and somatosensory conditions (modality-specific characteristic) because the distributions of RTs in those two conditions drew U-shapes. In addition, the U-shape distribution in those two conditions could have lead smaller IQRRT in PDEXP than in PDFLIP (modality-independent characteristic). If those suppositions were true, it would not be meaningful to separate modality-specific and -independent characteristics.

I am curious if the authors think that the RTs on the left and right side of the U-shapes (i.e., RTs in the sub and supra-second time range) can be explained by the model presented in this study. Previous studies have suggested that different neural mechanisms are involved in estimation of sub- and supra-second time intervals (for review: Buhushi and Meck, Nat. Rev. Neurosci 2005). I wish the authors add a discussion about how this study is consistent or inconsistent with the previous studies.

Yoshiko Yabe

Reviewer #3 (Remarks to the Author):

This study investigated the effect of the probability that an event will occur on the reaction times, in three sensory modalities: vision, audition and somatosensory. The authors demonstrate that the data can be best fit with the reciprocal of the probability density function in all three modalities, as opposed to the hazard rate of elapsed time.

There are several problems with the manuscript in its present form. First, the authors present two hypotheses about the computations related to the estimation of event occurrence, i.e. the hazard rate and the increasing uncertainty about elapsed time. However, these two hypotheses are not mutually exclusive. The probability distribution itself is blurred because of the scalar property of time perception, which also leads to a blurred hazard rate. It is misleading to present these two mechanisms as two alternative hypotheses.

Secondly, there was no time pressure at all in these experiments, since reward did not depend on reaction time. Therefore, we do not even know to what extent the participants were attending to elapsed time. RT changed over time, but there was no penalty for being slow and no reward for being fast.

I found the distinction between probabilistic and deterministic context confusing. It would be much cleaner to use the PDFs as they are, without truncating any part (i.e. 'deterministic context'). I believe we do not know exactly what the effect is of cutting the tail of the PDF on the participants' perception of the probability of an event. Furthermore, there was no spatial component at all in the task (merely a button press), and this makes the study difficult to compare to previous studies, since the neural representation of elapsed time is frequently reflected in motor planning activity.

Finally and most importantly, the authors make several claims that are not supported at all by the data. This is a purely behavioral study, therefore I cannot see how their findings 'represent a significant contribution towards a mechanistic understanding of long-standing questions' (p. 13).

Also, on p. 16 last paragraph the authors state that 'the Gaussian part of the ex-Gaussian model represents modality-specific processes of a peripheral nature, (...), and also shared motor processes'. But there is no evidence at all in the data for this conjecture.

Overall, this study only contributes new data about behavior related to the event probability over time in a very specific context: without time pressure, no spatial component to the response, truncated PDFs, etc.

Reviewers' Comments:

Reviewer #1:

Remarks to the Author:

Thank you for the thorough effort to address my concerns. The manuscript has been significantly improved. Although I still feel the proposed model is built upon somewhat arbitrary assumptions, I am supportive of publication. I think the authors have done an excellent job making their case. I anticipate that the model will be controversial and would therefore, provide the means for driving the field forward.

Reviewer #2:

Remarks to the Author:

The authors improved the manuscript very much. The authors responded to reviewers' comments well overall though I have two comments.

#1

They could still improve the Introduction to avoid a misunderstanding that the theme of this study is to judge whether hypothesis A or B is correct as the Reviewer #3 has suggested.

#2

Page 21 line 577-578

The authors' suggested that the modality-specific RT modulation at short 'go' times we observed may be unrelated to elapsed time estimation itself.

Does the elapsed time estimation include the tasks in reward-based context? This comment is just for my curiosity. So, the authors do not need to respond to this question in the manuscript at all but if the discussion about the elapsed time estimation could be somehow linked to the discussion on page 18, please add some discussion.

Minor comments

Figure 6a & b

The lines of somatosensory data are magenta but the lines in the legends are purple.

Figure 7a

Please check the y axis label of the som. graph.

Reviewer #3:

Remarks to the Author:

I have carefully read the reply by the authors. The manuscript has improved significantly and therefore I have no further comments. I am curious to see the response of the other reviewers.

We again thank the reviewers for their comments. We respond in detail, interleaving our replies in blue.

REVIEWER 1

Reviewer 1

Reviewer: Thank you for the thorough effort to address my concerns. The manuscript has been significantly improved. Although I still feel the proposed model is built upon somewhat arbitrary assumptions, I am supportive of publication. I think the authors have done an excellent job making their case. I anticipate that the model will be controversial and would therefore, provide the means for driving the field forward.

Reply:

We would like to thank the reviewer for recognizing our efforts to address her/his very valid and probing concerns and we feel delighted for her/his support of publication of this work. We are looking forward, with our current and future work, to contributing to the better understanding of how the brain learns and estimates probability in different domains.

REVIEWER 2

Reviewer: The authors improved the manuscript very much. The authors responded to reviewers' comments well overall though I have two comments.

#1 They could still improve the Introduction to avoid a misunderstanding that the theme of this study is to judge whether hypothesis A or B is correct as the Reviewer #3 has suggested.

Reply:

We thank the reviewer for pointing out this potential source of misunderstanding. We added a sentence in the introduction (highlighted in yellow) literally stating that hypotheses A and B are not tested against each other but are rather evaluated as model components in the present work.

Reviewer: #2 Page 21 line 577-578

The authors' suggested that the modality-specific RT modulation at short 'go' times we observed may be unrelated to elapsed time estimation itself.

Does the elapsed time estimation include the tasks in reward-based context? This comment is just for my curiosity. So, the authors do not need to respond to this question in the manuscript at all but if the discussion about the elapsed time estimation could be somehow linked to the discussion on page 18, please add some discussion.

Reply:

The reviewer correctly states that we hypothesize that the modality-specific RT modulation at short 'go' times we observed might be unrelated to elapsed time estimation itself. The reviewer asks whether the literature cited in this context includes rewarded tasks. In the section on elapsed time estimation that the reviewer cites, we hypothesize that it is potentially an effect of probability that modulates elapsed time estimation. Most of the work cited on this point is based on experiments with human participants in a non-rewarded context. Motivating effects of expected reward may very well also influence basic timing behavior, such as elapsed time estimation, in the context of temporal-probabilistic inference. However, we refrained from adding this line of thought to our discussion as it would take us far from the focus of our paper and, frankly, from the data we present in this work and the experimental setting that produced it (we manipulated event probability density, not reward). Nonetheless, we feel that this is an important question that requires targeted experiments itself. Given the constraints on the target length of the paper it is clearly beyond the scope of this work to discuss the potential effects of reward on elapsed time estimation adequately. Instead, we added a sentence (highlighted in yellow) to our hypothesis on how reward might modulate anticipation to clarify that anticipatory processes include both estimation of event probability and estimation of elapsed time and each process may be modulated by reward, linking it to the discussion on modality-specific RT modulation.

Minor comments

Reviewer: Figure 6a & b The lines of somatosensory data are magenta but the lines in the legends are purple.

Reply:

The lines in the legends were corrected, now matching the color of the data.

Reviewer: Figure 7a Please check the y axis label of the som. graph.

Reply:

We thank the reviewer for spotting this mistake. We have corrected the y axis labeling.

REVIEWER 3

Reviewer: I have carefully read the reply by the authors. The manuscript has improved significantly and therefore I have no further comments. I am curious to see the response of the other reviewers.

Reply:

We feel delighted that the reviewer's concerns have been addressed and that she/he is supporting the publication of our work.

ORIGINAL REVIEWERS' COMMENTS:

Reviewer #1 (Remarks to the Author):

Thank you for the thorough effort to address my concerns. The manuscript has been significantly improved. Although I still feel the proposed model is built upon somewhat arbitrary assumptions, I am supportive of publication. I think the authors have done an excellent job making their case. I anticipate that the model will be controversial and would therefore, provide the means for driving the field forward.

Reviewer #2 (Remarks to the Author):

The authors improved the manuscript very much. The authors responded to reviewers' comments well overall though I have two comments.

#1

They could still improve the Introduction to avoid a misunderstanding that the theme of this study is to judge whether hypothesis A or B is correct as the Reviewer #3 has suggested.

#2

Page 21 line 577-578

The authors' suggested that the modality-specific RT modulation at short 'go' times we observed may be unrelated to elapsed time estimation itself.

Does the elapsed time estimation include the tasks in reward-based context? This comment is just for my curiosity. So, the authors do not need to respond to this question in the manuscript at all but if the discussion about the elapsed time estimation could be somehow linked to the discussion on page 18, please add some discussion.

Minor comments

Figure 6a & b

The lines of somatosensory data are magenta but the lines in the legends are purple.

Figure 7a

Please check the y axis label of the som. graph.

Reviewer #3 (Remarks to the Author):

I have carefully read the reply by the authors. The manuscript has improved significantly and therefore I have no further comments. I am curious to see the response of the other reviewers.